# What Makes Value Learning Efficient in Residual Reinforcement Learning?

Guozheng Ma[1]  Lu Li[2 3]  Haoyu Wang[1]  Zixuan Liu[4]  Pierre-Luc Bacon[2 3]  Dacheng Tao[1]

## Abstract

Residual reinforcement learning (RL) enables stable online refinement of expressive pretrained policies by freezing the base and learning only bounded corrections. However, value learning in residual RL poses unique challenges that remain poorly understood. In this work, we identify two key bottlenecks: *cold start pathology*, where the critic lacks knowledge of the value landscape around the base policy, and *structural scale mismatch*, where the residual contribution is dwarfed by the base action. Through systematic investigation, we uncover the mechanisms underlying these bottlenecks, revealing that simple yet principled solutions suffice: base-policy transitions serve as an essential value anchor for implicit warmup, and critic normalization effectively restores representation sensitivity for discerning value differences. Based on these insights, we propose **DAWN** (Data-Anchored Warmup and Normalization), a minimal approach targeting efficient value learning in residual RL. By addressing these bottlenecks, **DAWN** demonstrates substantial efficiency gains across diverse benchmarks, policy architectures, and observation modalities. The code of **DAWN** is publicly available at GitHub ⊙.

## 1. Introduction

Generative policies pretrained on large-scale demonstrations, from Diffusion Policy (Chi et al., 2025) to Vision-Language-Action (VLA) models (Black et al., 2024; Intelligence et al., 2025), have reshaped the landscape of robot learning. Despite their expressiveness, imitation alone cannot meet the demands of reliable deployment due to inevitable distribution shift and compounding errors (Pan et al., 2026; Li et al., 2025). Simply scaling demonstration data cannot resolve this dilemma, since only interaction

[1]Nanyang Technical University [2]Mila - Quebec AI Institute [3]Université de Montréal [4]National University of Singapore. Correspondence to: Dacheng Tao <dacheng.tao@ntu.edu.sg>.

*Proceedings of the 43rd International Conference on Machine Learning*, Seoul, South Korea. PMLR 306, 2026. Copyright 2026 by the author(s).

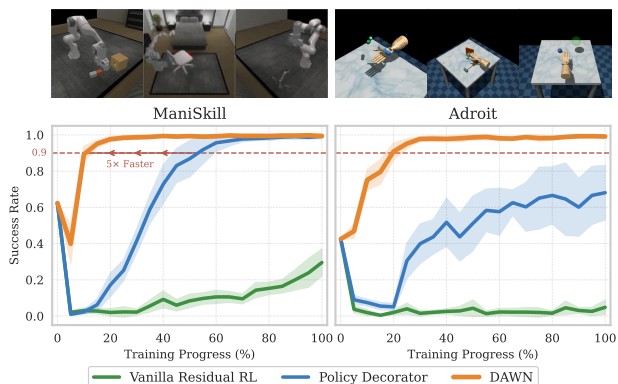

*Figure 1.* **DAWN enables efficient value learning in residual RL.** Aggregated success rates with Diffusion Policy as base policy across ManiSkill (3 tasks) and Adroit (3 tasks) benchmarks. **DAWN** achieves comparable final performance while converging approximately 5× faster than prior methods.

equips policies with the ability to recover through trial-and-error (Silver & Sutton, 2025). This makes **online reinforcement learning (RL)** essential for last-mile reliability, particularly in precision-critical domains such as contact-rich manipulation (Lei et al., 2025).

Among available approaches, residual RL offers a principled path for online refinement (Silver et al., 2018; Johannink et al., 2019): freezing the base policy and learning only a bounded correction avoids the forgetting and instability of modifying pretrained parameters (Jiang et al., 2024; Zhou et al., 2025). Recent work has advanced the paradigm through policy-centric innovations, from foundational techniques like bounded residuals (Ankile et al., 2025b) and progressive exploration (Yuan et al., 2025), to finer-grained designs such as per-step corrections (Ankile et al., 2025a) and latent steering for mode selection in multimodal policies (Anonymous, 2026). These efforts address the central question of *how to design an effective and stable residual policy*, making the paradigm practical for real-world refinement (Wang et al., 2025; Xiao et al., 2025).

However, effective policy design alone does not guarantee sample efficiency. Even state-of-the-art methods like Yuan et al. (2025) require over a million online interactions, a sample complexity prohibitive for real-world deployment. We argue that this gap stems from an overlooked factor: *value learning*. In the off-policy actor-critic methods under-

lying residual RL, policy improvement depends entirely on critic gradients (Haarnoja et al., 2018). If value learning is inefficient, no amount of policy engineering can compensate. This motivates three core research questions:

> **RQ1:** What are the unique and critical challenges for value learning in residual RL?
>
> **RQ2:** What bottlenecks underlie these challenges, and how can they be fundamentally resolved?
>
> **RQ3:** What gains in sample efficiency can be achieved through principled solutions?

Our investigation reveals two challenges unique to value learning in residual RL, each addressed by a simple but principled solution. • **Cold Start Pathology**: The critic begins without knowledge of the value landscape around the base policy. We find that transitions from the base policy serve as a necessary and sufficient *value anchor*, enabling effective implicit warmup. Explicit critic pre-training, by contrast, fails: freezing the policy during pre-training drives the value target into an entropy-dominated regime that corrupts value learning (Section 3.1). • **Structural Scale Mismatch**: The residual contribution is dwarfed by the base action, making it difficult for the critic to discern value differences. Normalization addresses this by restoring *critic sensitivity*, enabling effective credit assignment to residual actions. Despite input-level suppression, the residual's effect on Q-values manifests as a clear mean shift, making distributional objectives unnecessary (Section 3.2). By understanding these bottlenecks, the above insights naturally translate into our method, **DAWN** (Data-Anchored Warmup and Normalization). Focusing on value learning in residual RL, **DAWN** substantially improves sample efficiency, as illustrated in Figure 1. In summary, our contributions are:

1. We provide the first investigation of value learning in residual RL, identifying two unique challenges: *cold start pathology* and *structural scale mismatch*.

2. We uncover the mechanisms underlying these challenges through extensive analysis, revealing that warmup data provides an essential value anchor and normalization effectively restores critic sensitivity.

3. We propose **DAWN**, a principled yet minimal approach that is easy to implement. Extensive experiments across diverse benchmarks, policy architectures, and observation modalities demonstrate substantial improvements in sample efficiency.

## 2. Preliminary: Residual RL

Residual RL is built on a simple idea: instead of full policy optimization, learn only a small correction on top of an already capable base policy. This reduces the problem

from policy search to local refinement around a reliable prior. The paradigm originated from augmenting hand-crafted controllers with learned corrections (Silver et al., 2018; Johannink et al., 2019), and has since evolved to handle expressive policies trained through imitation. With the emergence of expressive policy models such as Diffusion Policy (Chi et al., 2025) and Vision-Language-Action models (Kim et al., 2024; Black et al., 2024), residual RL becomes particularly appealing for precision-critical tasks such as robotic manipulation. It enables stable online refinement of powerful offline-trained policies while avoiding the prohibitive sample complexity of learning from scratch.

**Problem Formulation.** In residual RL, the executed action combines a frozen base policy with a learned residual:

$$a = a_{\text{base}} + \lambda \cdot a_{\text{res}} = \underbrace{\pi_{\text{base}}(s)}_{\substack{\text{Pre-Trained} \\ \text{Frozen}}} + \lambda \cdot \underbrace{\pi_{\text{res}}(s)}_{\substack{\text{Randomly Initialized} \\ \text{Learnable}}} \quad (1)$$

where $\pi_{\text{base}}$ is typically obtained through imitation learning and remains fixed during RL training, while $\pi_{\text{res}}$ is a lightweight network (e.g., a small MLP) that learns task-specific corrections. The scaling factor $\lambda$ bounds the residual's influence and is typically set to a small value such as $0.1$. The critic takes the summed action as input, i.e., $Q(s, \pi_{\text{base}}(s) + \lambda \cdot \pi_{\text{res}}(s))$. This formulation, as opposed to concatenating base and residual actions separately, has been empirically validated as the most effective choice (Yuan et al., 2025; Ankile et al., 2025a). However, the small $\lambda$ means that the residual's contribution is dwarfed by the base action, creating a *structural scale mismatch* that makes it difficult for the critic to distinguish value differences caused by $\pi_{\text{res}}$, a challenge we analyze in Section 3.2.

A second challenge is *value cold start*. As an imitation-to-online method, residual RL must learn the value function from scratch since no offline data is available for critic pre-training. What makes residual RL distinct is that the base policy remains frozen, and all adaptation must proceed through a randomly initialized residual $\pi_{\text{res}}$. We examine how this structure affects value learning in Section 3.1.

**A Minimal Baseline for Investigation.** To isolate the core challenges of value learning, we adopt a minimal residual RL setup built on Soft Actor-Critic (SAC) (Haarnoja et al., 2018), which we use to train both the residual policy $\pi_{\text{res}}$ and the critic. SAC maintains two value networks $Q_{\theta_1}, Q_{\theta_2}$ with EMA-updated targets $Q_{\theta'_1}, Q_{\theta'_2}$, minimizing $\mathbb{E}[(Q_{\theta_i}(s,a) - y)^2]$ with the soft Bellman target:

$$y = r + \gamma \left( \min_{i \in \{1,2\}} Q_{\theta'_i}(s', a') - \alpha \log \pi(a' \mid s') \right), \quad (2)$$

where $\gamma$ is the discount factor and $\alpha$ the entropy temperature. The entropy term encourages exploration during normal training, but interacts non-trivially with the frozen

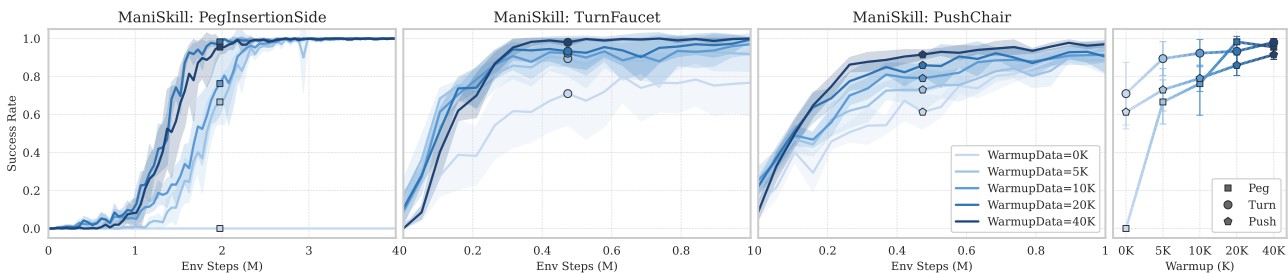

*Figure 2.* **Effect of warmup data quantity on learning performance.** (Left three) Learning curves across three ManiSkill tasks with varying amounts of warmup data. (Right) Success rate at the midpoint of training versus warmup data quantity. More warmup data consistently improves sample efficiency, with the effect most pronounced on challenging tasks. All experiments use 8 random seeds with shaded regions indicating 95% confidence intervals, a convention we follow throughout the paper.

residual policy during warmup, an effect we analyze in Section 3.1.2. The residual scale $\lambda$ (Equation 1) is the main task-specific hyperparameter, set to the values from Yuan et al. (2025); we deliberately avoid additional stabilization mechanisms to expose the raw dynamics of value learning. Full implementation details are in Appendix B.3.

## 3. Dissecting Value Learning in Residual RL

In residual RL, policy improvement relies entirely on critic gradients, making accurate value learning the foundation of efficient adaptation. To unlock this potential, this section dissects two challenges unique to value learning in residual RL: the cold start pathology (Section 3.1) and the structural scale mismatch (Section 3.2). For each challenge, we examine two candidate solutions motivated by different intuitions and analyze their effectiveness. Our investigation demonstrates that effectively addressing these challenges substantially improves sample efficiency, while providing insights into the underlying mechanisms and the principles behind effective solutions. All experiments use the minimal baseline described in Section 2 on ManiSkill (Gu et al., 2023) tasks with Diffusion Policy (Chi et al., 2025) as the base policy; full details are provided in Appendix.

### 3.1. Challenge I: The Cold Start Pathology

At the start of training, the critic is randomly initialized and has no knowledge of the value landscape around the base policy. In online RL from scratch, value estimates and policy co-evolve from random initialization, allowing the critic to mature alongside policy improvements. In residual RL, however, the full policy operates near a competent base policy from the outset. If the critic cannot quickly ground itself to this region, erroneous value estimates will misguide the residual policy, risking catastrophic performance collapse.

Two intuitions suggest potential remedies. ● The first perspective emphasizes *data as implicit warmup*: effective value learning may require sufficient warmup transitions before training begins. ● The second emphasizes *explicit warmup training*: perhaps the critic requires dedicated pre-training before the residual policy starts updating. We inves-

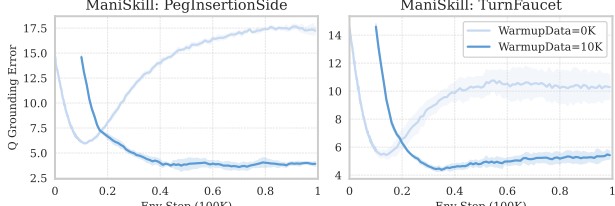

*Figure 3.* **Q-value grounding error during early training.** Without warmup data, the error briefly decreases but quickly diverges. With warmup, the critic maintains accurate estimates throughout, confirming the value anchor effect.

tigate both perspectives below. We first investigate the role of warmup data and find that on-policy data from the base policy alone serves as an effective value anchor, enabling efficient value learning from the start (Section 3.1.1). We then examine whether explicit critic pre-training offers additional benefits, and find that freezing the policy for pre-training is instead counterproductive (Section 3.1.2).

#### 3.1.1. THE NECESSITY OF WARMUP DATA AS ANCHOR

We begin by examining how the amount of warmup data affects learning. Before online training, we collect transitions using the base policy and store them in the replay buffer. The critic then learns from this data alongside newly collected transitions once training starts.

**Effect on Learning Performance.** Figure 2 shows that warmup data substantially affects learning efficiency. Without warmup, the most challenging task (PegInsertionSide) fails entirely, while easier tasks suffer significant sample inefficiency. Performance improves consistently as more warmup transitions are provided. Based on these results, we use 20K warmup transitions as default in subsequent experiments. This finding aligns with concurrent work (Ankile et al., 2025a), which shows that incorporating offline demonstrations into the replay buffer improves stability in residual RL. Our result extends this insight: rather than requiring pre-collected demonstrations with reward labels, simply collecting transitions from the base policy is sufficient.

**The Value Anchor Effect.** Seeding the replay buffer with initial transitions is standard practice in off-policy RL (Lillicrap et al., 2015). However, with a random policy, such

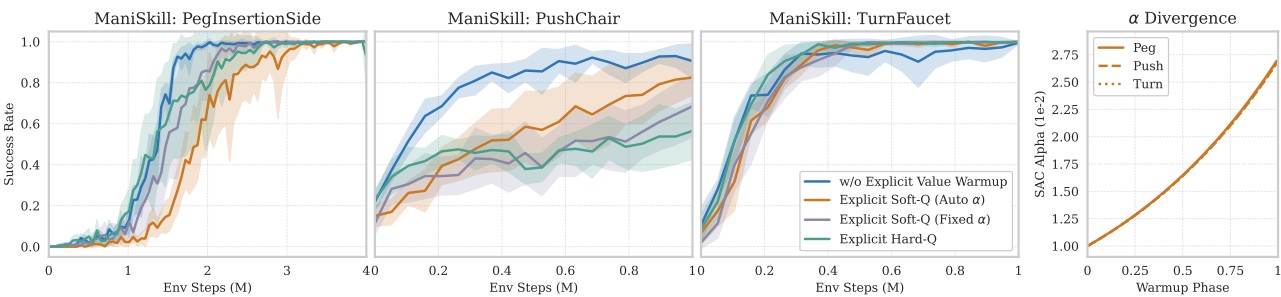

*Figure 4.* **Effect of explicit value warmup on learning performance.** (Left three) Explicit warmup variants fail to improve and often degrade sample efficiency compared to implicit warmup alone. (Right) With automatic entropy tuning, $\alpha$ diverges during the warmup phase across all tasks, even with an initial value as small as $0.01$. Larger initial values lead to more severe divergence (see Appendix).

data carries limited information and is typically kept minimal. In residual RL, the situation differs: warmup data collected from the base policy provides meaningful trajectories as a prior for value learning. Specifically, since residual RL refines the base policy through local corrections rather than global policy search, the critic must accurately capture the value landscape in the neighborhood of the base policy. Warmup data serves as a *value anchor* that grounds the critic to this region before residual learning begins.

To verify this anchor effect, we measure the *Q-value grounding error*: the discrepancy between the critic's estimates and the true returns on base policy trajectories. Concretely, we compute $\mathcal{E}_{\text{grounding}} = |Q_\theta(s, a_{\text{base}}) - G^{\pi_{\text{base}}}|$, where $G^{\pi_{\text{base}}}$ is the Monte Carlo return. This metric directly measures how well the critic captures the value landscape in the region where residual learning operates. Details on the experimental design and formal justification are in Appendix C.1.2. Figure 3 reveals a clear contrast: • Without warmup data, the grounding error briefly decreases but quickly diverges, indicating the critic fails to establish stable value estimates. • With warmup data, the error drops rapidly and remains low, confirming that warmup data successfully anchors the critic to the relevant value landscape.

A natural question is whether adding exploration during warmup would improve coverage and accelerate learning. We compare the base policy alone against several exploration strategies (details in Appendix C.1.1). Surprisingly, the base policy alone matches or exceeds all alternatives. This reinforces the value anchor interpretation: grounding the critic matters more than collecting diverse trajectories. Exploration emerges naturally once training begins through the stochastic policy and entropy regularization.

> **Takeaway:** Warmup data addresses cold start by serving as a **value anchor**: providing a prior over the value landscape around the base policy is both necessary and sufficient for bootstrapping efficient value learning.

### 3.1.2. THE REDUNDANCY OF EXPLICIT WARMUP PHASE

Given that warmup data provides an effective value anchor, a natural question arises: ***can explicit critic pre-training***

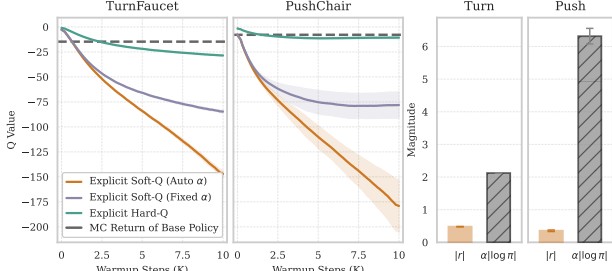

*Figure 5.* **The failure mechanism of explicit value warmup.** (Left) Q-value estimates during the warmup phase. Soft Q methods collapse to extreme negative values, while Hard Q remains near the true MC return. (Right) During explicit Soft Q warmup, the magnitude of $|\alpha \log \pi|$ substantially exceeds $|r|$.

***further accelerate learning?*** We investigate this by freezing the residual policy and training only the critic during an initial warmup phase. We consider three representative variants that span the main design choices: • *Explicit Soft Q (Auto $\alpha$)*: standard SAC critic updates with automatic entropy tuning; • *Explicit Soft Q (Fixed $\alpha$)*: critic updates with entropy coefficient fixed at its initial value; and • *Explicit Hard Q*: critic updates using the standard Bellman target without entropy regularization. Contrary to expectations, Figure 4 shows that all explicit warmup variants fail to improve sample efficiency and often make it worse. We analyze the failure modes of each approach below.

**Entropy Dominance and Value Collapse.** Explicit Soft Q warmup fails under both entropy settings, for a common reason that is intrinsic to freezing the policy rather than to the entropy schedule. With autotuned $\alpha$, the temperature diverges during warmup (Figure 4), as also observed in Yuan et al. (2025); with $\alpha$ fixed, the value estimates instead collapse to extreme negative values, far below the true Monte Carlo returns (Figure 5, left). Both stem from *entropy dominance*: when $|\alpha \log \pi|$ substantially exceeds $|r|$ (Figure 5, right), the entropy term governs the TD target and injects a large negative bias that the critic fits in place of the true value landscape; in the autotuned case, the temperature update compounds this by raising $\alpha$ in a futile attempt to reach the target entropy.

Critically, the large $|\log \pi|$ driving this effect is not an artifact of how $\pi_{\text{res}}$ is initialized. Under the standard SAC

log-std parameterization, any zero-mean symmetric initialization places the expected log-std at the midpoint of its admissible range, deep in the near-deterministic regime where $|\log \pi|$ is large; since this midpoint is independent of the initialization scale, widening the initial variance does not escape it (Appendix C.2). Action chunking, characteristic of the expressive policies we refine, further inflates the effective action dimension over which $|\log \pi|$ accumulates.

The decisive factor is therefore not the entropy term itself but the frozen policy. In ordinary joint training, the first actor updates move the policy off the midpoint almost immediately, so $|\log \pi|$ shrinks and the bias is transient; explicit warmup keeps $\pi_{\text{res}}$ fixed throughout pre-training, leaving the critic no such escape. This is why explicit Soft Q warmup fails where standard SAC does not.

**Why Hard Q Warmup Is Not the Solution.** A natural response is to drop the entropy term that drives the collapse. Hard Q indeed avoids it, fitting values close to the true Monte Carlo returns during warmup (Figure 5, left). The difficulty instead surfaces at the switch to Soft Q training. The soft value landscape is systematically more negative than the hard one, because every Bellman backup carries the entropy penalty, which accumulates over the effective horizon. The pre-trained critic must therefore adjust from one value scale to a substantially different one. This is not the kind of bias that joint training corrects on its own: there the entropy term shrinks as the actor moves off its initial regime, whereas here the entire target range shifts at once and the critic must relearn it from scratch. The switch therefore costs more than simply starting from random initialization with proper warmup data.

> **Takeaway:** Explicit value warmup is **_redundant_** in residual RL. Freezing $\pi_{\text{res}}$ traps the policy in a near-deterministic regime where entropy dominates the TD target and corrupts value learning, while removing entropy instead creates an objective mismatch. The implicit warmup through data anchoring alone is **_both simpler and more effective_** for addressing the cold start pathology.

## 3.2. Challenge II: The Structural Scale Mismatch

The bounded residual formulation (Equation 1) uses a small $\lambda$ to ensure that residual corrections remain local (Yuan et al., 2025). However, this introduces a structural challenge for value learning that is unique to residual RL.

Unlike standard RL where the entire action is learnable, residual RL introduces a structural asymmetry: the frozen base action dominates in magnitude, while the learnable residual is suppressed by a factor of $\lambda$. When this asymmetric combination $a = a_{\text{base}} + \lambda \cdot a_{\text{res}}$ serves as input to the critic, the resulting *scale mismatch* poses a fundamental

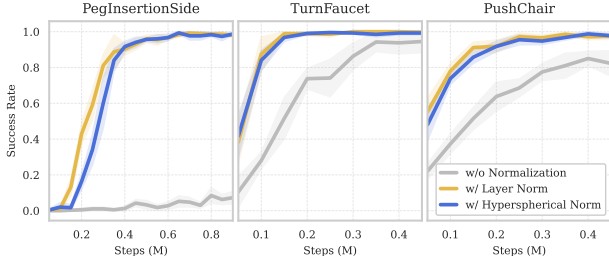

*Figure 6.* **Effect of critic normalization.** Both LN and HN substantially improve learning efficiency compared to the unnormalized baseline, with comparable performance to each other.

challenge for credit assignment. To guide policy improvement, the critic must distinguish value differences caused by different residual actions. However, $a_{\text{base}}$ dominates the input magnitude, making it difficult for the critic to correctly attribute value to the residual action.

To address this scale mismatch, we investigate two methodological paths. ● The first focuses on *architectural intervention*: employing normalization techniques to decouple the critic's internal representations from input magnitudes, allowing it to remain sensitive to small variations regardless of the dominant base action. ● The second considers *objective refinement*: distributional RL provides richer learning signals by modeling the full return distribution, potentially capturing value distinctions that mean-based critics miss.

### 3.2.1. RESTORING SENSITIVITY VIA NORMALIZATION

Given the scale mismatch identified above, we explore whether normalization techniques can restore the critic's sensitivity to residual actions. We examine two representative approaches: Layer Normalization (LN) (Ba et al., 2016) and Hyperspherical Normalization (HN) (Lee et al., 2025b).

LN normalizes activations across features within each layer:

$$\text{LN}(\mathbf{x}) = \gamma \odot \frac{\mathbf{x} - \mu}{\sigma} + \beta \qquad (3)$$

where $\mu$ and $\sigma$ are the mean and standard deviation computed over the feature dimension, and $\gamma$ and $\beta$ are learnable affine parameters. This stabilizes the activation statistics regardless of input magnitude.

Unlike LN which normalizes statistical moments (Lee et al., 2025a), HN (Lee et al., 2025b) enforces a strict geometric constraint. For a layer with input $\mathbf{x} \in \mathbb{R}^d$ and weight $\mathbf{w} \in \mathbb{R}^d$, both are projected onto the unit hypersphere:

$$\hat{\mathbf{x}} = \mathbf{x}/\|\mathbf{x}\|_2, \quad \hat{\mathbf{w}} = \mathbf{w}/\|\mathbf{w}\|_2 \qquad (4)$$

The pre-activation output then depends solely on the angle $\theta$ between vectors, independent of their original magnitudes:

$$y = s \cdot (\hat{\mathbf{w}}^T \hat{\mathbf{x}}) = s \cdot \cos(\theta) \qquad (5)$$

where $s$ is a learnable scalar. By explicitly removing magnitude information, HN ensures the critic responds only to directional changes in the input.

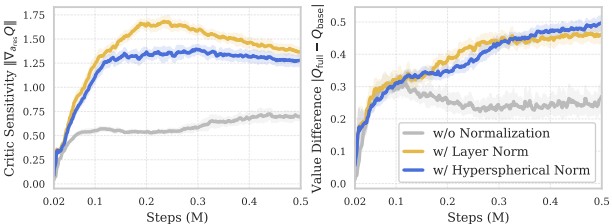

*Figure 7.* **Mechanism analysis of normalization.** *(Left)* Critic sensitivity to residual actions, measured by $\|\nabla_{a_{\text{res}}}Q\|$. Normalization yields significantly higher sensitivity, enabling the critic to detect small residual variations. *(Right)* Value contribution of the residual policy, measured by $|Q_{\text{full}} - Q_{\text{base}}|$. With normalization, the critic learns to attribute increasing value to the residual throughout training. *Evaluated on PegInsertionSide.*

**Effect on Learning Efficiency.** Building on the warmup data foundation, we evaluate whether normalization can address the scale mismatch challenge. Figure 6 compares sample efficiency with and without normalization applied to the critic network. Both LN and HN substantially improve sample efficiency across all tasks, with the most pronounced gains on the PegInsertionSide task. Notably, although LN and HN operate through different mechanisms (statistical normalization versus geometric projection), they achieve comparable performance. This suggests that the key factor is restoring sensitivity to residual variations, not the specific normalization strategy.

**Mechanism: How Normalization Helps.** To understand why normalization improves learning, we analyze two diagnostic metrics on the PegInsertionSide task (Figure 7).

The left panel shows the critic's sensitivity to residual actions, measured by $\|\nabla_{a_{\text{res}}}Q\|$. This gradient norm quantifies how strongly the critic's output responds to changes in the residual. With normalization, sensitivity rises rapidly and reaches approximately twice the level of the unnormalized baseline, confirming that normalization enables the critic to better detect residual variations despite their small magnitude. The right panel shows the estimated value contribution of the residual policy, measured by $|Q(s, a_{\text{base}} + \lambda \cdot a_{\text{res}}) - Q(s, a_{\text{base}})|$. This quantity captures the value difference the critic assigns to the residual action. With normalization, this contribution grows steadily throughout training, indicating that the critic learns to recognize the residual's effect on value. Without normalization, growth is substantially slower, suggesting the critic struggles to distinguish the residual's contribution.

**Robustness to Residual Scale.** The residual scale $\lambda$ controls the magnitude of policy corrections and may vary across tasks. We compare LN and HN across a range of $\lambda$ values to assess their robustness. As shown in Figure 8, while both methods perform well at moderate scales, HN degrades significantly at extreme values such as $\lambda = 0.05$. In contrast, LN maintains stable performance across the entire range. Although HN has demonstrated strong performance

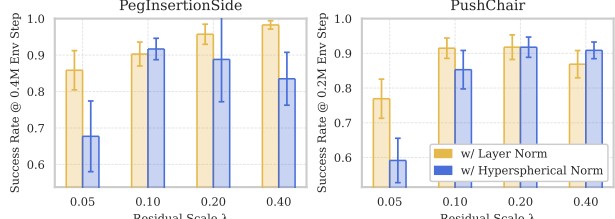

*Figure 8.* **Robustness to residual scale $\lambda$.** LN maintains stable performance across all $\lambda$ values, while HN exhibits higher variance and degrades significantly at small scales.

in settings involving network scaling or massively parallel training (Lee et al., 2025b; Seo et al., 2025), its strict normalization may be less suited to the fine-grained adaptation required in residual RL. In this setting, LN's adaptive statistics provide greater flexibility. Combined with its simplicity, we adopt LN as the default for critic normalization.

> **Takeaway:** Normalization effectively addresses the scale mismatch. Both statistical (LN) and geometric (HN) approaches restore critic sensitivity to residual actions, substantially improving value learning efficiency.

### 3.2.2. Revisiting Distributional Objectives

Distributional RL models the full return distribution $Z(s, a)$ rather than just its expectation $Q(s, a) = \mathbb{E}[Z(s, a)]$, providing richer learning signals that have proven beneficial in large-scale and multi-task scenarios (Lee et al., 2025b; Nauman et al., 2025). Given the scale mismatch challenge, where small residual variations must be distinguished in the value landscape, we investigate whether this additional expressiveness offers advantages over scalar regression.

We compare the standard MSE objective against three distributional alternatives, all implemented within the SAC framework by replacing the scalar critic with distributional variants: • *C51* (Bellemare et al., 2017): models returns as a categorical distribution over a fixed set of atoms, trained with cross-entropy loss. • *QR* (Dabney et al., 2018): learns a set of return quantiles via quantile regression, without requiring fixed support. • *TQC* (Kuznetsov et al., 2020): extends quantile regression by truncating the highest estimates to mitigate overestimation. While these methods differ in distributional representation, they share the goal of capturing richer information beyond the mean. Full implementation details are provided in Appendix.

**Distributional Objectives Offer No Efficiency Gains.** Building on the warmup and normalization foundations established above, all tasks converge to near-perfect success rates. To assess whether distributional objectives provide additional benefits, we compare early-stage performance on two challenging tasks. As shown in Figure 9, MSE achieves competitive or superior efficiency compared to all distributional variants, indicating that the added complexity of

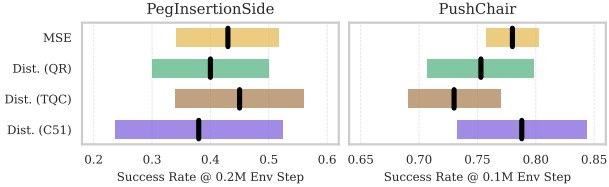

*Figure 9.* **Comparison of critic objectives.** MSE achieves competitive or superior early-stage performance compared to distributional methods (C51, QR, TQC), indicating no significant efficiency gains from modeling the full return distribution.

modeling the full return distribution yields no significant improvement for residual value learning.

**Why MSE Suffices.** Figure 10 provides mechanistic insight into why distributional objectives offer no advantage. At a critical training checkpoint, we sample 1024 state-action pairs and visualize both the action distributions and the corresponding Q-value distributions produced by the critic.

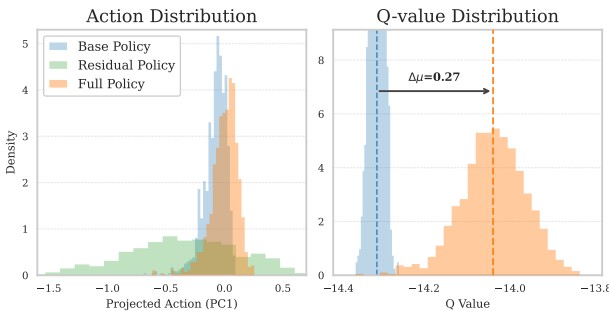

*Figure 10.* **The anatomy of residual value learning.** At a training checkpoint, we visualize action and Q-value distributions from 1024 samples. *(Left)* Actions projected onto the first principal component. The residual induces a subtle shift, illustrating scale mismatch at the input level. *(Right)* Despite the small action shift, Q-value distributions show clear separation ($\Delta\mu = 0.27$), confirming that the residual's effect manifests as a distinct mean shift. *Evaluated on PegInsertionSide with LN.*

The left panel projects actions onto their first principal component. The residual *shifts the action distribution only slightly*, so the full policy stays close to the base in action space. This is exactly the scale mismatch challenge for value learning: because the residual is scaled by a small $\lambda$, it makes up only a small part of the combined action that the critic sees, even though it is the only part the critic must learn to evaluate. The right panel shows the corresponding output: despite the marginal action shift, the *Q-value distributions exhibit clear separation*. Normalization enables the critic to translate subtle input differences into a distinct value signal. Importantly, this signal manifests as a pronounced *mean shift* rather than a nuanced change in distribution shape. Since MSE directly targets mean differences, it is well-suited to capture the value changes induced by residual policies. In contrast, distributional methods must learn the entire distribution shape, a strictly harder objective that offers no efficiency advantage when the mean alone provides a sufficient learning signal.

**Takeaway:** Although the residual occupies only a small part of the critic's action input, its effect on the Q-values shows up as a pronounced mean shift. This makes MSE a natural fit for residual value learning, with distributional objectives offering no additional efficiency benefit.

## 4. `DAWN`: Unlocking Efficient Value Learning

Efficient value learning in residual RL requires addressing two fundamental challenges: the *cold start pathology* and the *structural scale mismatch*. We propose `DAWN` (**D**ata-**A**nchored **W**armup and **N**ormalization), which targets each challenge with a minimal, principled intervention:

> **Data-Anchored Warmup** $\Rightarrow$ *Cold Start Pathology*
> Before training, collect $N_{\text{warmup}}$ transitions using the base policy alone and store them in the buffer.
>
> **Critic Normalization** $\Rightarrow$ *Structural Scale Mismatch*
> Apply Layer Normalization to the critic network.

These two interventions are complementary and sufficient. Data-anchored warmup ensures the critic begins with accurate value estimates in the relevant region, while normalization enables it to distinguish the subtle variations introduced by the residual policy. Together, they establish the foundation for efficient value learning without requiring modifications to the underlying RL algorithm.

The power of `DAWN` lies not in novel components. Warmup data collection and normalization are both well-established techniques; our contribution lies in identifying *why* and *where* they become essential for residual value learning. Complex alternatives such as carefully tuned distributional objectives, task-specific warmup data collection strategies, or elaborate exploration schedules may yield marginal gains, but they do not address the core bottlenecks. In contrast, the simplicity of `DAWN` is a direct consequence of targeting the fundamental challenges: once the critic is properly anchored and sensitive to residual variations, efficient value learning follows naturally. This simplicity also yields strong generality and robustness: beyond the residual scale $\lambda$ inherent to any bounded residual formulation, `DAWN` introduces only a single hyperparameter, the number of warmup transitions $N_{\text{warmup}}$, yet demonstrates consistent effectiveness across diverse tasks, base policies, and observation modalities.

## 5. Experiments

We assess whether insights from Section 3 yield practical gains by evaluating `DAWN` across diverse settings.

**Setup.** We consider two benchmarks: ManiSkill (Gu et al., 2023) for contact-rich manipulation and Adroit (Rajeswaran et al., 2017) for dexterous hand control. To show that our

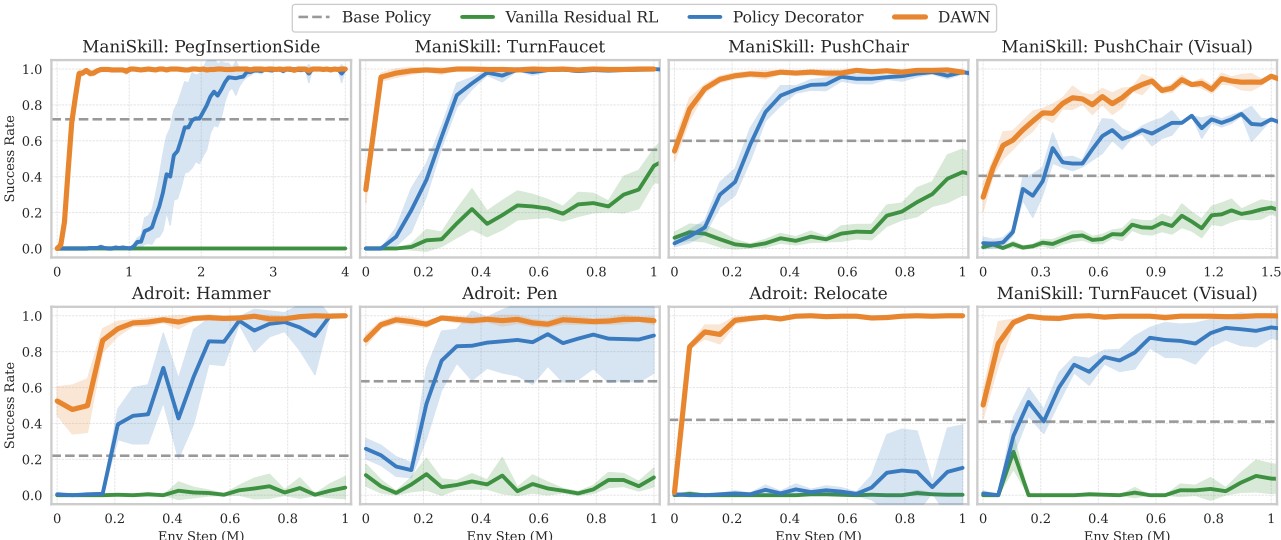

*Figure 11.* **Main results with Diffusion Policy as the base policy.** Across 8 tasks spanning ManiSkill and Adroit benchmarks (including 2 with visual observations), **DAWN** consistently achieves superior sample efficiency compared to all baselines, converging to high success rates significantly faster. Dashed lines indicate base policy performance. Shaded regions denote 95% confidence intervals over 8 seeds.

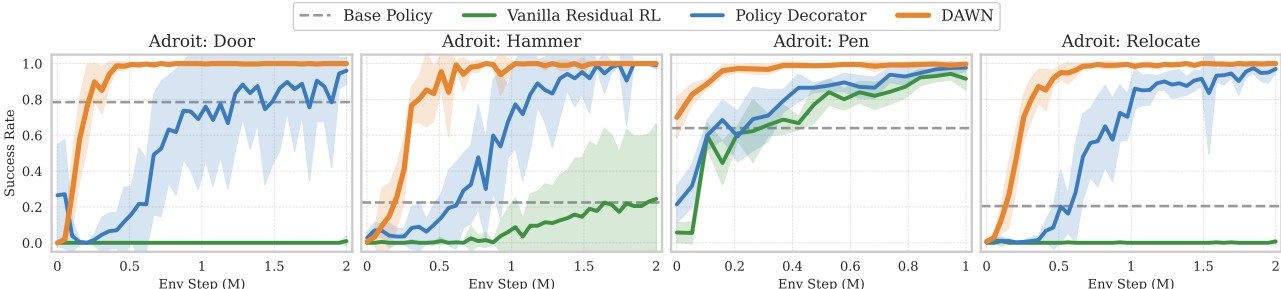

*Figure 12.* **Main results with BeT as the base policy.** **DAWN** maintains strong performance across all Adroit tasks, demonstrating robustness to different base policy architectures. Shaded regions show 95% confidence intervals over 8 seeds.

insights apply beyond a specific base policy, we consider two distinct architectures: Diffusion Policy (Chi et al., 2025) and Behavior Transformer (BeT) (Shafiullah et al., 2022). Full implementation details are provided in Appendix E.

**Baselines.** Our primary baseline is *Policy Decorator* (Yuan et al., 2025), the state-of-the-art residual RL method that introduces bounded residual actions and progressive exploration for stable learning. We also include *Vanilla Residual RL* (Johannink et al., 2019) as a minimal reference without stabilization mechanisms. Since Policy Decorator has been shown to substantially outperform methods such as JSRL (Uchendu et al., 2023), Cal-QL (Nakamoto et al., 2023), and FISH (Haldar et al., 2023), we focus our comparison on this stronger baseline.

**Main Results.** Figure 11 presents results with Diffusion Policy as the base policy across 8 tasks spanning ManiSkill and Adroit benchmarks, including 2 tasks with visual observations; Figure 12 further validates generalization to BeT as the base policy. **DAWN** consistently achieves superior sample efficiency compared to all baselines, converging to high success rates significantly faster. On the challenging

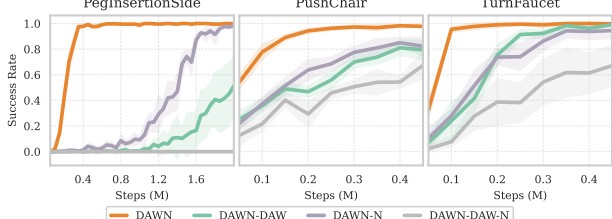

*Figure 13.* **Ablation on DAWN components.** Data-anchored warmup (**DAW**) and critic normalization (**N**) address complementary bottlenecks. Removing either component degrades performance.

PegInsertionSide task, **DAWN** reaches 90% success approximately $6\times$ faster than Policy Decorator, demonstrating that directly addressing the fundamental bottlenecks of value learning yields far greater efficiency gains than defensive mechanisms like progressive exploration. Beyond efficiency, **DAWN** exhibits notably low variance across seeds on all tasks, suggesting that addressing the root causes of inefficient value learning also improves training stability. Such reliability is essential for real-world deployment.

**Ablation.** Figure 13 confirms that both components of **DAWN** are essential: removing either data-anchored warmup

or critic normalization substantially degrades learning efficiency. As analyzed in Section 3, these two components target distinct bottlenecks and their benefits do not overlap. We further examine whether normalization benefits the actor, the critic, or both. Adding normalization to the actor provides no improvement and may slightly degrade performance (Figure 14), consistent with our analysis that scale mismatch is primarily a value-side challenge.

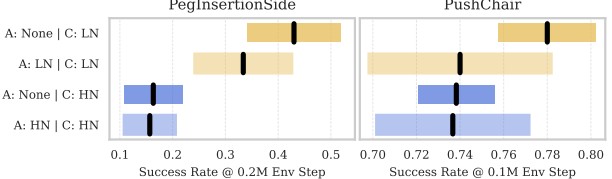

*Figure 14.* **Effect of actor *vs.* critic normalization.** Normalization benefits the critic but not the actor, confirming that scale mismatch affects value learning rather than policy optimization.

Beyond these component studies, we verify that DAWN's gains are robust along three axes. ● First, against stronger baselines: DAWN matches or exceeds ResFiT (Ankile et al., 2025a) while requiring only base-policy rollouts rather than a reward-annotated offline dataset, and remains competitive with diffusion-policy fine-tuning methods such as DSRL (Wagenmaker et al., 2025) and DPPO (Ren et al., 2024) (Appendix F.1). ● Second, beyond SAC: instantiating DAWN in the DDPG-based ResFiT framework preserves both the gains and the necessity of each component, confirming that cold start and scale mismatch stem from the residual formulation rather than from SAC's entropy mechanism (Appendix F.2). ● Third, against compute scaling: higher update-to-data ratios and larger critic ensembles cannot substitute for resolving the underlying value-learning pathologies (Appendix F.3). Together these confirm that DAWN's improvements come from addressing the right bottlenecks, not from incidental implementation choices. Further ablations on hyperparameter sensitivity and design variations are provided in Appendix G.

## 6. Related Work

Residual RL refines a fixed base policy by learning a bounded correction, and prior work has improved it almost entirely through policy-side mechanisms: action bounding, progressive exploration, and controlled exploration strategies (Yuan et al., 2025; Ankile et al., 2025a; Xu et al., 2025). These treat the symptoms of unstable training rather than its cause. We instead study residual RL from a value-learning perspective, and show that cold start and scale mismatch are the underlying bottlenecks. Crucially, the two techniques we use are not new; our contribution is identifying *why* and *where* they become essential in the residual setting, which prior policy-centric work left unexamined.

The closest prior work uses warmup or normalization in

adjacent settings. In offline-to-online fine-tuning, warmup transitions recalibrate an *already pre-trained* Q-function across distribution shift (Zhou et al., 2025); in residual RL the critic is randomly initialized and must be anchored to the base policy's value landscape from scratch, a distinct problem. Concurrently, ResFiT (Ankile et al., 2025a) applies critic normalization (following RLPD) on real-world humanoid systems, but relies on a reward-annotated offline dataset and applies normalization to the full network; we show that base-policy rollouts alone suffice as a value anchor and that the effect is specific to the critic. A full discussion of offline-to-online RL, foundation-policy fine-tuning, and normalization in deep RL is deferred to Appendix A.

## 7. Conclusion

This work investigated what makes value learning a bottleneck in residual RL and how to address it. By identifying *cold start pathology* and *scale mismatch* as the root causes, we developed **DAWN**, a minimal approach that achieves substantial efficiency gains through two simple interventions: data-anchored warmup and critic normalization.

Beyond proposing a specific algorithm, this work provides mechanistic insight into value learning dynamics unique to residual RL. For instance, we show that despite the residual's small footprint in action space, its effect on Q-values manifests as a clear mean shift, explaining both why standard critics struggle to assign credit and why capturing the full value distribution is unnecessary. More broadly, this work illustrates a research philosophy: rather than introducing complex machinery to mask symptoms, tracing problems to their root causes enables solutions that are both simpler and more effective.

**Limitations and Future Work.** Our study has several limitations, which also point to its most natural extensions. First, our evaluation is entirely in simulation; although **DAWN** leaves the underlying algorithm untouched and relies only on base-policy rollouts, confirming these benefits on real hardware, where resets are costly and sample budgets are tight, is an important next step. Second, the base policies we refine are relatively compact, and whether our conclusions carry over to large vision-language-action models, whose scale and action parameterization differ substantially, remains open. Third, although our benchmarks span contact-rich and dexterous manipulation, **DAWN** solves most within a moderate budget, leaving value learning on substantially harder or longer-horizon tasks not yet fully characterized. We emphasize that these reflect the limits of our empirical scope rather than of the identified bottlenecks, which stem from the residual formulation itself and are therefore expected to generalize. We hope this work encourages the community to consider the critical role of value learning in fine-tuning foundation policies.

# Acknowledgment

This project is supported by the National Research Foundation, Singapore, under its NRF Professorship Award No. NRF-P2024-001.

# Impact Statement

This paper presents work whose goal is to advance the field of Machine Learning. There are many potential societal consequences of our work, none which we feel must be specifically highlighted here.

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

# Appendix for

## *What Makes Value Learning Efficient in Residual Reinforcement Learning?*

## Part I: Background and Setup

## Part II: Investigation Details

## Part III: Method and Ablations

# A. Extended Related Work

## A.1. Residual Reinforcement Learning

Residual RL learns a correction term on top of a fixed base policy, preserving prior knowledge while adapting to unmodeled dynamics. Originally applied to classical controllers in contact-rich robotics tasks (Silver et al., 2018; Johannink et al., 2019; Haldar et al., 2023; Yan et al., 2026), the paradigm has recently gained traction for online refinement of large pretrained policies, including behavior cloning (Ankile et al., 2025b), diffusion policies (Anonymous, 2026), and vision-language-action models (Xiao et al., 2025), where direct fine-tuning is sample-inefficient and risks catastrophic forgetting.

Prior work has addressed training challenges primarily through policy-side mechanisms: action bounding, progressive exploration schedules, and controlled exploration strategies (Silver et al., 2018; Yuan et al., 2025; Ankile et al., 2025a; Xu et al., 2025). These techniques treat symptoms of instability rather than root causes. In contrast, we study residual RL from a *value learning* perspective, identifying critic cold start and scale mismatch as the fundamental bottlenecks, and develop principled fixes that address these causes directly.

## A.2. Fine-tuning Foundation Policies via RL

Large pretrained policies such as diffusion policies (Chi et al., 2025), behavior transformers (Shafiullah et al., 2022), and vision-language-action models (Zitkovich et al., 2023) have become powerful foundations for robot learning but often underperform in deployment due to distribution shift. RL fine-tuning offers a natural remedy, yet existing approaches face distinct challenges depending on whether they modify the base policy.

Direct fine-tuning methods (Ren et al., 2024; Zhang et al., 2025; Mark et al., 2024) update the base policy parameters via policy gradients. While effective, these approaches must carefully manage training stability, catastrophic forgetting, and architecture-specific optimization challenges. In contrast, residual methods (Yuan et al., 2025; Ankile et al., 2025b;a; Wagenmaker et al., 2025) freeze the base policy and learn only a lightweight correction, offering two key advantages: *stability* by preserving the base policy's knowledge, and *model-agnosticism* by applying regardless of base architecture. A separate line of work from the offline-to-online RL literature (Nakamoto et al., 2023; Ball et al., 2023; Uchendu et al., 2023) accelerates fine-tuning by leveraging offline data to initialize critics or guide exploration. However, these methods typically assume access to offline datasets with reward annotations or require specific algorithmic machinery (e.g., conservative Q-learning), limiting their applicability when only a pretrained policy is available.

Our work uses residual RL as a controlled setting to study value learning dynamics. We show that targeted improvements to the critic can substantially boost efficiency, suggesting that a deeper understanding of RL optimization fundamentals may complement ongoing efforts in policy fine-tuning.

## A.3. Normalization in Deep RL

Normalization has become a key ingredient for stable and scalable value learning, along two threads. Statistical normalization, popularized by RLPD (Ball et al., 2023) and extended by SimBa (Lee et al., 2025a) and high-capacity value functions (Nauman et al., 2025; Ma et al., 2025), inserts Layer Normalization (Ba et al., 2016) into the critic to curb value extrapolation and enable scaling of width and update-to-data ratios. Geometric normalization, such as Hyperspherical Normalization (Lee et al., 2025b) and FlashSAC (Kim et al., 2026), projects weights and activations onto the unit sphere to keep representations well-conditioned under massive parallelism.

What unites these advances is that they target capacity and throughput, addressing a bottleneck of *scale* within the network. The residual setting poses a different problem: the critic need not be large, yet it must stay sensitive to the $\lambda$-scaled residual that is structurally suppressed by the dominant base action. Normalization helps precisely because it decouples internal representations from input magnitude, restoring this sensitivity (Section 3.2.1), and the decoupling matters more than the specific mechanism, as statistical and geometric variants both succeed.

They differ mainly in robustness. Hyperspherical normalization discards magnitude entirely, which suits the throughput scaling it was designed for but is less forgiving of fine-grained adaptation, whereas Layer Normalization's adaptive statistics remain stable across the full range of residual scales we test (Figure 8). More broadly, while normalization is standard in from-scratch and large-scale RL, its role in restoring critic sensitivity under a fixed, dominant base action had not been examined; our contribution is to identify why and where it becomes essential in this regime.

# B. Experimental Setup

## B.1. Environment Details

We evaluate on several tasks from two complementary benchmarks that together cover a broad spectrum of manipulation challenges: ManiSkill (Mu et al., 2021; Gu et al., 2023) for contact-rich manipulation, and Adroit (Rajeswaran et al., 2017) for high-dimensional dexterous control with an anthropomorphic hand.

**ManiSkill.** ManiSkill is a large-scale benchmark for generalizable manipulation skills built on the SAPIEN simulator (Xiang et al., 2020). It provides diverse task families with object-level geometric and topological variations, making it well-suited for studying generalization in manipulation. We select three challenging tasks that stress different aspects of residual learning:

- **PegInsertionSide**: Insert a peg into a horizontal hole with randomized box geometry. This task demands sub-millimeter precision and is highly sensitive to policy errors, making it an ideal testbed for measuring the efficiency of value learning.

- **TurnFaucet**: Rotate a faucet handle to turn it on. The base policy is trained on 10 faucet instances, while online evaluation uses 4 *unseen* faucets, testing generalization to novel object geometries.

- **PushChair**: A dual-arm mobile manipulator must push a swivel chair to a target location without tipping it over. Trained on 5 chair instances and evaluated on 3 unseen chairs with randomized friction and damping parameters.

All ManiSkill tasks use a Franka Panda robot (or dual-arm variant for PushChair) with delta end-effector control. For visual observation experiments, we use $64 \times 64$ RGBD images from base and hand-mounted cameras.

**Adroit.** The Adroit benchmark (Rajeswaran et al., 2017) features the Shadow Dexterous Hand, a 24-DoF anthropomorphic hand mounted on a 4–6 DoF arm, totaling 28–30 DoF depending on the task. Originally introduced to study demonstration-augmented policy gradient (DAPG), these tasks remain challenging benchmarks for dexterous manipulation due to their high-dimensional action spaces and contact-rich dynamics. We evaluate on four tasks:

- **Door**: Undo a latch with significant dry friction and bias torque, then swing the door open. The door position is randomized at each episode.

- **Pen**: Reorient a pen to match a randomized target configuration, requiring precise in-hand manipulation.

- **Hammer**: Grasp a hammer and drive a nail into a board. The nail position is randomized and exerts up to 15N of resistive friction.

- **Relocate**: Move a ball to a target location, with both positions randomized across the workspace.

All Adroit tasks use absolute joint position control.

**Task Specifications.** Table 1 summarizes the observation and action dimensions for each task. The high action dimensionality of Adroit tasks (24–30 DoF) presents a distinct challenge compared to the lower-dimensional but precision-critical ManiSkill tasks.

*Table 1.* Task specifications across benchmarks. ManiSkill tasks feature lower action dimensions but require high precision; Adroit tasks involve high-dimensional dexterous control.

| Task | State Dim | Action Dim | Episode Length |
|------|-----------|------------|----------------|
| ManiSkill: PegInsertionSide | 50 | 7 | 200 |
| ManiSkill: TurnFaucet | 43 | 7 | 200 |
| ManiSkill: PushChair | 131 | 20 | 200 |
| Adroit: Door | 39 | 28 | 300 |
| Adroit: Pen | 46 | 24 | 200 |
| Adroit: Hammer | 46 | 26 | 400 |
| Adroit: Relocate | 39 | 30 | 400 |

## B.2. Base Policy Training

***To ensure fair comparison with prior work, we use the pretrained base policy checkpoints released by** Yuan et al. (2025)* ***rather than training our own.*** This section documents the base policy architectures and training procedures for reference; readers primarily interested in our method can skip to Section B.3.

We experiment with two expressive policy architectures: Diffusion Policy (Chi et al., 2025) and Behavior Transformer (BeT) (Shafiullah et al., 2022). Diffusion Policy models the action distribution as a conditional denoising process and has demonstrated strong performance on contact-rich manipulation tasks. BeT uses a transformer architecture with action discretization via k-means clustering, enabling multimodal action prediction.

**Diffusion Policy.** We use the U-Net variant of Diffusion Policy. Key architecture hyperparameters are listed in Table 2. The policy observes 2 timesteps of history and predicts 16 future actions, of which 4 are executed before re-planning.

*Table 2.* Diffusion Policy architecture.

| Hyperparameter | Value |
| --- | --- |
| Observation horizon | 2 |
| Prediction horizon | 16 |
| Action horizon | 4 |
| Embedding dimension | 64 |
| Down-sampling dimensions | 256, 512, 1024 |
| Parameters | ∼4M |

**Behavior Transformer.** BeT discretizes the action space using k-means clustering and models action sequences autoregressively. Architecture hyperparameters are listed in Table 3.

*Table 3.* Behavior Transformer architecture.

| Hyperparameter | Value |
| --- | --- |
| Context window | 10 / 20 |
| Number of clusters | 4 / 8 |
| Transformer layers | 4 |
| Attention heads | 4 |
| Embedding dimension | 128 |
| Parameters | ∼1M |

**Training.** Both policies are trained using the AdamW optimizer with a learning rate of $10^{-4}$. Diffusion Policy is trained for 200K gradient steps with batch size 1024 on all tasks. BeT is trained for 200K steps on ManiSkill tasks and 5K steps on Adroit tasks (which have fewer demonstrations), both with batch size 2048. Checkpoints are selected based on the highest evaluation success rate over 50 episodes, evaluated every 50K steps.

**Demonstration Data.** Table 4 summarizes the demonstration data used for base policy training. ManiSkill demonstrations are generated via motion planning or model predictive control and contain 1000 trajectories per task. Adroit demonstrations are collected via human teleoperation with only 25 trajectories per task, reflecting the original DAPG setup (Rajeswaran et al., 2017).

*Table 4.* Demonstration data for base policy training.

| Benchmark | Trajectories | Collection Method |
| --- | --- | --- |
| ManiSkill | 1000 | Motion planning / MPC |
| Adroit | 25 | Human teleoperation |

### B.3. Details of Minimal Baseline

Our implementation builds on **Policy Decorator** (Yuan et al., 2025), which introduces two strategies for stable residual learning. The first is *bounded residual action*: scaling the residual output by a small $\lambda$ (as in Equation 1) to prevent large deviations from the base policy. The second is *progressive exploration*: gradually increasing the probability of using the residual policy during training,

$$\pi_{\text{behavior}}(s) = \begin{cases} \pi_{\text{base}}(s) + \lambda \cdot \pi_{\text{res}}(s) & \text{if } u < \epsilon \\ \pi_{\text{base}}(s) & \text{otherwise} \end{cases} \tag{6}$$

where $u \sim \text{Uniform}(0,1)$ and $\epsilon = \min(t/H, 1)$ increases linearly until timestep $H$, a task-specific hyperparameter that varies widely across tasks.

We adopt bounded residual action with the task-specific $\lambda$ values from Yuan et al. (2025). However, we deliberately omit progressive exploration for two reasons:

1. As a protective mechanism, progressive exploration masks the symptoms of inefficient value learning, obscuring the dynamics we aim to study.

2. Once value learning is efficient, such protection becomes unnecessary and potentially harmful: the changing $\epsilon$ introduces training non-stationarity, and the mixture of base-only and full-policy rollouts creates distribution mismatch between data collection and value estimation.

This minimal setup exposes the raw challenges of value learning and serves as our experimental subject in Section 3.

**Off-Policy RL Algorithm.** We use Soft Actor-Critic (SAC) (Haarnoja et al., 2018) as the underlying off-policy algorithm, following the standard practice in residual RL (Yuan et al., 2025; Ankile et al., 2025a). SAC maintains two value networks $Q_{\theta_1}, Q_{\theta_2}$ along with EMA-updated target networks $Q_{\theta'_1}, Q_{\theta'_2}$, optimized by minimizing $\mathbb{E}[(Q_{\theta_i}(s,a) - y)^2]$ with the soft Bellman target:

$$y = r + \gamma \left( \min_{i \in \{1,2\}} Q_{\theta'_i}(s', a') - \alpha \log \pi(a' \mid s') \right), \tag{7}$$

where $\gamma$ is the discount factor and $\alpha$ is the temperature for entropy regularization. The entropy term encourages exploration during normal training, but interacts non-trivially with the *frozen* residual policy during explicit warmup. As we analyze in Section 3.1.2, freezing $\pi_{\text{res}}$ keeps it in a near-deterministic regime where $|\log \pi|$ remains large regardless of the initialization scale, causing the entropy term to dominate the TD target and corrupt value learning.

**Network Architecture.** The residual policy $\pi_{\text{res}}$ is a lightweight MLP with 2 hidden layers of 256 units each, using ReLU activations. The critic $Q_\theta$ follows the same architecture but takes the state together with the *summed action* $a = a_{\text{base}} + \lambda \cdot a_{\text{res}}$ as input. We use twin critics to mitigate overestimation bias. To avoid disrupting the base policy at initialization, the **residual policy's final layer is initialized with near-zero weights**, producing near-zero actions with minimal variance at the start of training.

*Table 5.* SAC hyperparameters.

| Hyperparameter | ManiSkill | Adroit |
|---|---|---|
| Discount $\gamma$ | 0.97 / 0.9 | 0.97 |
| Batch size | 1024 | 1024 |
| Learning rate | $10^{-4}$ | $10^{-4}$ |
| Initial entropy $\alpha$ | 0.01 | 0.01 |
| Policy update frequency | 1 | 1 |
| Env steps per update | 64 | 64 |
| Update-to-data ratio | 0.25 | 0.25 |
| Target network $\tau$ | 0.01 | 0.01 |
| Replay buffer size | $10^6$ | $10^6$ |

*Table 6.* Residual scale $\lambda$ for investigation tasks (Section 3).

| Task | $\lambda$ |
|---|---|
| PegInsertionSide | 0.1 |
| TurnFaucet | 0.1 |
| PushChair | 0.2 |

**Hyperparameters.**    Table 5 lists the SAC hyperparameters shared across all experiments. Table 6 lists the residual scale $\lambda$ for the ManiSkill tasks used in Section 3; full $\lambda$ values for all tasks are provided in Section E. Note that standard off-policy RL implementations include a "learning starts" parameter that specifies the number of transitions to collect before training begins. The original Policy Decorator uses 8K transitions for this purpose. Our investigation in Section 3.1.1 reveals that this warmup data serves as a critical value anchor for residual RL, and we systematically study the effect of varying this quantity from 0K to 40K. Based on these findings, we adopt 20K warmup transitions as the default for **DAWN**.

## C. Investigation of Cold Start Pathology

### C.1. Implicit Warmup

This section provides additional details and analysis on the implicit warmup mechanism discussed in Section 3.1.1.

#### C.1.1. WARMUP DATA COLLECTION STRATEGY

Section 3.1.1 establishes that warmup data is necessary for efficient value learning. A natural follow-up question arises: *how should this warmup data be collected?* One might expect that adding exploration during warmup would improve state-action coverage and thereby accelerate subsequent learning. To test this hypothesis, we compare four collection strategies:

- **Base Policy Only**: Collect transitions using $\pi_{\text{base}}$ alone, without any exploration mechanism. This is the simplest strategy and the one we adopt by default.

- **Full Action**: Execute the full combined policy $\pi_{\text{base}} + \lambda \cdot \pi_{\text{res}}$ with the randomly initialized residual. Since $\pi_{\text{res}}$ outputs near-zero actions at initialization, this introduces only minimal perturbations to the base policy's behavior.

- **Gaussian Noise**: Add isotropic Gaussian noise to the base policy actions, i.e., $a = \pi_{\text{base}}(s) + \epsilon$ where $\epsilon \sim \mathcal{N}(0, \sigma^2 I)$. We set $\sigma = 0.1$ to match the typical residual scale $\lambda$.

- **Epsilon-Greedy**: Mix base-only and full-policy rollouts following Equation 6 with a fixed ratio $\epsilon = 0.2$, where 20% of transitions use the combined policy and 80% use the base policy alone.

**Results.**    Figure 15 compares these strategies across two ManiSkill tasks. Contrary to the intuition that more exploration should help, *Base Policy Only* performs as well as or better than all exploration-augmented strategies. The failure modes are task-dependent: Gaussian Noise causes PegInsertionSide to collapse entirely, while PushChair exhibits higher variance across strategies but remains more robust to perturbations.

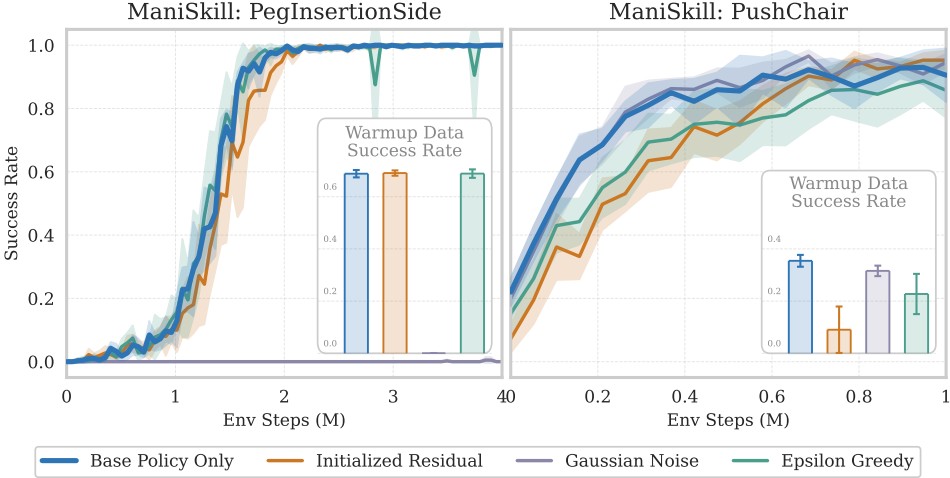

*Figure 15.* **Comparison of warmup data collection strategies.** *Base Policy Only* achieves comparable or better performance than strategies with additional exploration. The inset bar charts show success rates during warmup data collection; exploration degrades warmup performance in a task-dependent manner.

**Analysis.**   This result reinforces the *value anchor* interpretation of warmup data. The purpose of warmup is not to pre-collect diverse trajectories that cover a wide state-action space, but rather to ground the critic to accurate value estimates in the region where the base policy operates. Since residual RL refines the base policy through local corrections rather than global policy search, the critic must first understand the value landscape around $\pi_{\text{base}}$ before it can meaningfully guide residual learning. Adding exploration during warmup dilutes this grounding signal with transitions from regions that may be irrelevant or misleading for the initial phase of learning.

Once RL training begins, exploration emerges naturally through two mechanisms: (1) the stochastic residual policy $\pi_{\text{res}}$, which samples actions from a learned Gaussian distribution, and (2) the entropy regularization in SAC, which explicitly encourages exploration. TD learning then propagates value estimates outward from the anchored foundation established by warmup data. Our results do not rule out the possibility that more sophisticated exploration strategies could be beneficial; designing active exploration methods that specifically target value-critical regions for anchoring remains a promising direction for future work. Here, we demonstrate that the simplest approach—collecting data from the base policy alone—already provides substantial benefits and offers a clear mechanistic explanation.

**Trajectory-Consistent Noise: A Negative Result.**   The failure of step-wise Gaussian noise raises a natural question: is the problem with noise itself, or with its high-frequency nature? In contact-rich manipulation tasks, the dynamics exhibit significant inertia and friction. Step-wise white noise is largely filtered out by the physics, producing trajectories that stay close to the base behavior despite the added perturbations. When the noise magnitude is increased to overcome this filtering, the robot quickly diverges from reasonable behavior and the success rate drops to near zero.

To test whether *trajectory-consistent* noise could address this issue, we designed a more structured exploration strategy. Instead of sampling independent noise at each timestep, we sample a single bias vector $b \sim \mathcal{N}(0, \sigma^2 I)$ at the start of each episode and apply it consistently throughout:

$$a_t^{\text{res}} = b + \epsilon_t, \quad \text{where } \epsilon_t \sim \mathcal{N}(0, \sigma_{\text{jitter}}^2 I) \tag{8}$$

with $\sigma = 0.2$ for the trajectory-level bias and $\sigma_{\text{jitter}} = 0.01$ for small per-step jitter. This design ensures that the effect of the residual integrates over time, producing clear differences in observed returns rather than being filtered out by the dynamics.

We compare two variants:

- **TC Noise**: All trajectories use trajectory-consistent noise as described above.

- **TC Noise w/ Anchor**: Half of the trajectories use the pure base policy (anchor), and half use trajectory-consistent noise.

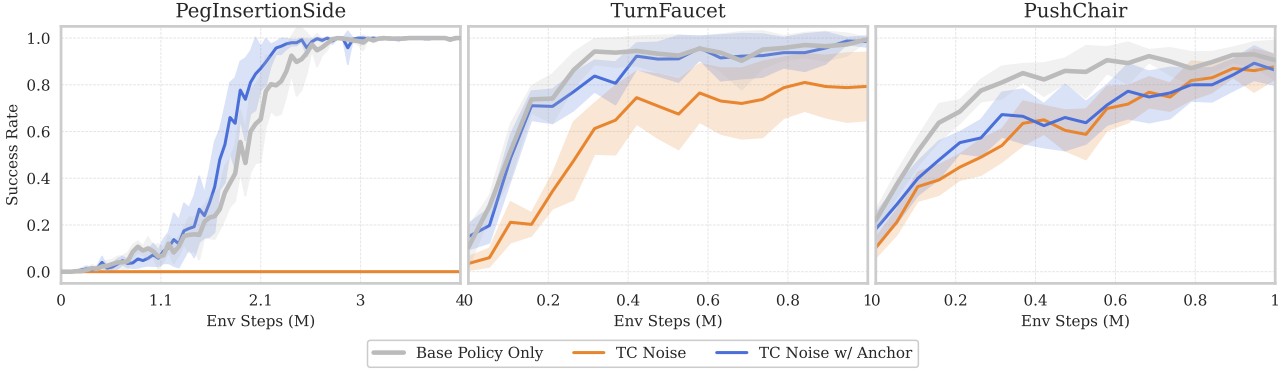

*Figure 16.* **Trajectory-consistent noise does not improve over base policy only.** TC Noise fails entirely on PegInsertionSide and underperforms on TurnFaucet. Adding anchor trajectories (TC Noise w/ Anchor) recovers performance but provides no benefit over the simpler Base Policy Only strategy.

Despite the more principled design, neither variant outperforms the simple *Base Policy Only* strategy:

- **TC Noise** performs substantially worse than *Base Policy Only*, confirming that exploration during warmup—even when carefully structured—dilutes the value anchor signal.

- **TC Noise w/ Anchor** performs comparably to or slightly worse than *Base Policy Only*. Adding anchor trajectories recovers most of the performance, but the probe trajectories provide no additional benefit.

This negative result reinforces our main finding: the purpose of warmup data is to anchor the critic to accurate values in the base policy's region, not to explore the state-action space. The probe trajectories, despite their structured design, introduce transitions from perturbed behaviors that are irrelevant or misleading for the initial phase of value learning. The simplest approach—collecting data from the base policy alone—remains the most effective.

### C.1.2. Q-VALUE GROUNDING ERROR

To verify the value anchor effect quantitatively, we measure the *Q-value grounding error*: the discrepancy between the critic's estimates and the true returns on base policy trajectories.

**Definition.** Let $\tau = (s_0, a_0, r_0, s_1, \ldots)$ be a trajectory collected by rolling out the base policy $\pi_{\text{base}}$. The Monte Carlo return from state $s_t$ is:

$$G_t^{\pi_{\text{base}}} = \sum_{k=0}^{T-t} \gamma^k r_{t+k} \tag{9}$$

The grounding error at timestep $t$ is:

$$\mathcal{E}_{\text{grounding}}(s_t, a_t) = |Q_\theta(s_t, a_t) - G_t^{\pi_{\text{base}}}| \tag{10}$$

We report the average grounding error over a held-out set of base policy trajectories collected before training begins.

**Justification I: Why Grounding Error over TD Error?** A natural alternative would be to track the temporal difference (TD) error during training. However, TD error measures *Bellman consistency*, i.e., how well the critic aligns with its own bootstrapped target, rather than alignment with the true value landscape. In the cold start phase, the target network $Q_{\theta'}$ is randomly initialized and unreliable. Consequently, a low TD error may merely indicate that the critic has learned to predict the output of another random network (*bootstrapping bias*), without learning the underlying task structure. In contrast, the Monte Carlo return $G^{\pi_{\text{base}}}$ provides an *unbiased* reference derived directly from environmental rewards, independent of any neural network estimates. By measuring the grounding error against this external reference, we verify whether the critic is genuinely anchored to the physical value landscape rather than merely self-consistent.

**Justification II: Why Base Policy Trajectories?** Our metric evaluates the critic on pure base policy trajectories, while the critic ultimately learns to evaluate the combined policy $\pi_{\text{base}} + \lambda \cdot \pi_{\text{res}}$. This approximation is justified in our setting for two reasons. First, at the start of training, $\pi_{\text{res}}$ is initialized to output near-zero actions, making the combined policy nearly identical to $\pi_{\text{base}}$. Second, accurate value estimates for $\pi_{\text{base}}$ are a prerequisite for guiding residual learning—the critic cannot improve upon a baseline it does not understand. The grounding error thus measures exactly what matters for cold start: whether the critic has learned the value landscape in the region where residual learning will operate.

**Results.** As shown in Figure 3 of the main text, the grounding error exhibits starkly different dynamics depending on whether warmup data is provided:

- **Without warmup data**: The error briefly decreases in the first few thousand steps but then diverges rapidly. This pattern suggests that the critic initially makes progress but, lacking a stable anchor, eventually overfits to spurious patterns and drifts away from the true value landscape.

- **With warmup data**: The error drops rapidly at the start of training and remains low throughout. The warmup transitions provide a stable foundation that grounds the critic to the relevant region before residual learning begins.

This quantitative analysis confirms that warmup data serves as a value anchor: it grounds the critic to the true value landscape around $\pi_{\text{base}}$, enabling meaningful policy improvement from the start of training.

### C.2. Explicit Warmup

This section provides additional details on the explicit warmup experiments discussed in Section 3.1.2.

C.2.1. EXPERIMENTAL PROTOCOL

We investigate whether explicit critic pre-training can further accelerate learning beyond the implicit warmup provided by data anchoring. During the explicit warmup phase, we freeze the residual policy $\pi_{\text{res}}$ and train only the critic for a fixed number of steps before joint actor-critic training begins.

**Setup.** We use 20K warmup transitions collected from the base policy (as established in Section 3.1.1). The explicit warmup phase consists of 10K gradient steps on the critic, corresponding to half of the warmup data being sampled on average per update. After the warmup phase, training proceeds normally with joint actor-critic updates.

**Variants.** We consider three representative variants that span the main design choices for critic pre-training:

• **Explicit Soft Q (Auto $\alpha$)**: Standard SAC critic updates with automatic entropy tuning. The critic is updated by minimizing:

$$\mathcal{L}_Q = \mathbb{E}_{(s,a,r,s')\sim\mathcal{D}}\left[\left(Q_\theta(s,a) - y\right)^2\right] \tag{11}$$

where the target $y$ includes the entropy term:

$$y = r + \gamma\left(\min_{i\in\{1,2\}} Q_{\theta'_i}(s',a') - \alpha\log\pi(a'\mid s')\right), \quad a' \sim \pi(\cdot\mid s') \tag{12}$$

The entropy coefficient $\alpha$ is updated automatically to maintain a target entropy level:

$$\mathcal{L}_\alpha = \mathbb{E}_{a\sim\pi}\left[-\alpha\left(\log\pi(a\mid s) + \bar{\mathcal{H}}\right)\right] \tag{13}$$

where $\bar{\mathcal{H}}$ is the target entropy (typically $-\dim(\mathcal{A})$).

• **Explicit Soft Q (Fixed $\alpha$)**: Same as above, but the entropy coefficient is fixed at its initial value ($\alpha = 0.01$) throughout the warmup phase, disabling automatic tuning. The target remains:

$$y = r + \gamma\left(\min_{i\in\{1,2\}} Q_{\theta'_i}(s',a') - \alpha\log\pi(a'\mid s')\right) \tag{14}$$

• **Explicit Hard Q**: Critic updates using the standard Bellman target without entropy regularization:

$$y = r + \gamma\min_{i\in\{1,2\}} Q_{\theta'_i}(s',a'), \quad a' \sim \pi(\cdot\mid s') \tag{15}$$

This removes the entropy term entirely during warmup, avoiding the entropy-related pathologies but introducing an objective mismatch with the subsequent Soft Q training.

In all variants, the residual policy $\pi_{\text{res}}$ remains frozen during the warmup phase, outputting near-zero actions with minimal variance due to its initialization.

C.2.2. WHY $|\log\pi|$ IS LARGE INDEPENDENT OF INITIALIZATION

We now make precise why the frozen residual policy sits in a regime where $|\log\pi|$ is large, and why this is a structural consequence of *freezing* rather than of how $\pi_{\text{res}}$ is initialized.

**Log-std parameterization.** SAC's stochastic actor outputs a state-dependent mean $\mu_\phi(s)$ and a raw log-std head $y_\phi(s)$. To keep the standard deviation in a stable range, standard implementations map $y$ onto a bounded interval $[L_{\min}, L_{\max}]$ through a $\tanh$ rescaling:

$$\log\sigma = L_{\min} + \tfrac{1}{2}(L_{\max} - L_{\min})\left(\tanh(y) + 1\right), \tag{16}$$

where we follow the standard bounds $L_{\min} = -20$, $L_{\max} = 2$, so the interval midpoint is $-9$.

**The expected log-std sits at the midpoint, independent of the initialization scale.** At initialization, the final-layer weights of the log-std head are drawn from a zero-mean symmetric distribution, so $y$ is approximately zero-mean and symmetric. Since $\tanh$ is an odd function, $\mathbb{E}[\tanh(y)] = 0$ for *any* symmetric distribution of $y$, regardless of its variance. Substituting into Equation 16,

$$\mathbb{E}[\log \sigma] = L_{\min} + \tfrac{1}{2}(L_{\max} - L_{\min}) = \tfrac{1}{2}(L_{\min} + L_{\max}) = -9. \tag{17}$$

This midpoint depends only on $(L_{\min}, L_{\max})$, not on the variance of the initialization. Widening the initialization variance of the log-std head only spreads $\tanh(y)$ toward the two extremes of $[L_{\min}, L_{\max}]$ while keeping $\mathbb{E}[\tanh y] = 0$ by symmetry; the policy therefore remains, on average, deep in the near-deterministic regime. This is why re-initializing $\pi_{\mathrm{res}}$ with larger variance (we tested up to $100\times$) does not escape it.

**Resulting magnitude of $|\log \pi|$.** For a $\tanh$-squashed Gaussian, the log-density of an action $a' = \tanh(u')$ with $u' \sim \mathcal{N}(\mu, \mathrm{diag}(\sigma^2))$ is

$$\log \pi(a' \mid s) = \sum_{i=1}^{d} \left[ -\tfrac{1}{2}\big((u'_i - \mu_i)/\sigma_i\big)^2 - \log \sigma_i - \tfrac{1}{2}\log(2\pi) - \log\big(1 - \tanh^2(u'_i)\big) \right]. \tag{18}$$

Taking the expectation over $u' \sim \pi$ gives $\mathbb{E}[((u'_i - \mu_i)/\sigma_i)^2] = 1$, and the $\tanh$ Jacobian term is negligible because the residual mean is near zero ($\tanh(\mu_i) \approx 0$). Per dimension this leaves

$$\mathbb{E}[\log \pi_i] \approx -\log \sigma_i - \tfrac{1}{2}\big(1 + \log 2\pi\big) \approx 9 - 1.42 = 7.58, \tag{19}$$

using $\mathbb{E}[\log \sigma_i] = -9$. The total log-density therefore scales with the effective action dimension $d$ as $\mathbb{E}[\log \pi] \approx 7.58\,d$.

**Amplification by action chunking.** The expressive policies we refine predict action chunks, so the effective dimension is $d = \dim(\mathcal{A}) \times H_{\mathrm{act}}$ where $H_{\mathrm{act}}$ is the action horizon. For PegInsertionSide ($\dim(\mathcal{A}) = 7$, $H_{\mathrm{act}} = 4$), $d = 28$ and $\mathbb{E}[\log \pi] \approx 7.58 \times 28 \approx 212$, matching the empirical value. Even with the small initial temperature $\alpha = 0.01$, the entropy term $\alpha |\log \pi| \approx 2.12$ substantially exceeds the reward magnitude $|r| \leq 1$, so the entropy term governs the TD target.

**Why freezing is the decisive factor.** In ordinary joint training, the first actor updates move $y_\phi$ off its symmetric initialization, so $\mathbb{E}[\tanh y]$ departs from zero, $\log \sigma$ rises, and $|\log \pi|$ shrinks within a few updates; the entropy bias is therefore transient. Explicit warmup keeps $\pi_{\mathrm{res}}$ frozen and removes exactly this escape route, trapping the critic in the near-deterministic regime throughout pre-training. The pathology is thus a structural consequence of freezing the policy, and adjusting the initial entropy coefficient changes only the rate of divergence, not its occurrence.

### C.2.3. EFFECT OF INITIAL ENTROPY COEFFICIENT

Section 3.1.2 shows that with automatic entropy tuning, $\alpha$ diverges during the explicit warmup phase. Here we examine how the initial value of $\alpha$ affects this divergence.

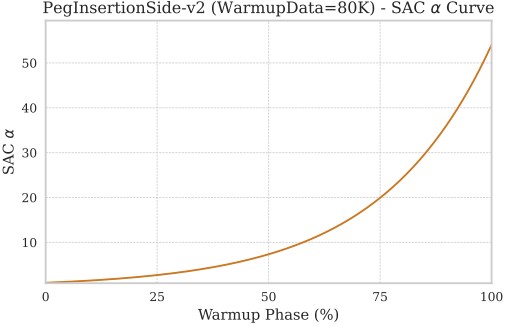

*Figure 17.* **Effect of initial entropy coefficient on $\alpha$ divergence.** Larger initial values lead to more severe divergence during explicit warmup. With $\alpha_{\mathrm{init}} = 1.0$, the entropy coefficient explodes rapidly, reaching values orders of magnitude higher than with $\alpha_{\mathrm{init}} = 0.01$.

Figure 17 compares the $\alpha$ dynamics during explicit warmup with different initial values. The divergence is consistently observed across all initial values, but larger initial $\alpha$ leads to more severe and rapid divergence. This is because the automatic tuning mechanism attempts to maintain a target entropy level. With $\pi_{\text{res}}$ frozen in the near-deterministic regime (where $|\log\pi|$ is large for reasons independent of the initialization scale; see below), the tuning rule continuously increases $\alpha$ in a futile attempt to reach the target entropy. The larger the initial $\alpha$, the larger the entropy term $|\alpha\log\pi|$ becomes, which further destabilizes the TD target and accelerates the divergence. This result confirms that the entropy dominance problem is fundamental to the explicit warmup setting and cannot be resolved by simply adjusting the initial entropy coefficient.

### C.2.4. EFFECT OF WARMUP DATA QUANTITY

One might hypothesize that explicit warmup fails because the warmup data (20K transitions) is insufficient for meaningful critic pre-training. To test this, we increase the warmup data to 80K transitions and extend the explicit warmup phase to 40K gradient steps on the PegInsertionSide task.

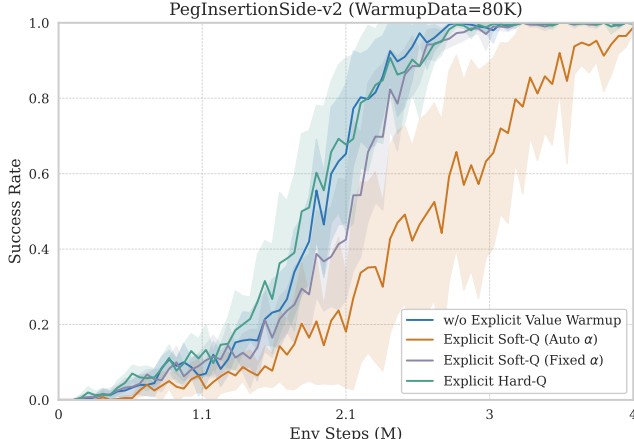

*Figure 18.* **Increasing warmup data does not salvage explicit warmup.** Even with $4\times$ more warmup data (80K transitions) and proportionally longer warmup phase (40K steps), explicit Soft Q warmup still fails to improve over implicit warmup alone.

As shown in Figure 18, increasing the quantity of warmup data does not resolve the failure of explicit warmup. The entropy dominance problem persists regardless of data quantity: with $\pi_{\text{res}}$ frozen throughout the warmup phase, $|\log\pi|$ remains large irrespective of the initialization scale, causing the entropy term to dominate the TD target and corrupt value learning. This confirms that the failure of explicit warmup is not a matter of insufficient data, but a fundamental incompatibility between the SAC entropy mechanism and a *frozen* residual policy.

## D. Investigation of Scale Mismatch

### D.1. Critic Normalization

This section provides additional analysis on the normalization experiments in Section 3.2.1.

### D.1.1. THE NECESSITY OF NORMALIZATION

The main text compares Layer Normalization (LN) and Hyperspherical Normalization (HN). Here we include the unnormalized baseline to emphasize that *normalization itself* is the critical factor, while the choice between LN and HN is secondary. Figure 19 shows performance across different residual scales $\lambda \in \{0.05, 0.1, 0.2, 0.4\}$. The unnormalized critic fails almost entirely on PegInsertionSide regardless of $\lambda$, and performs substantially worse on PushChair. In contrast, both LN and HN achieve strong performance across the range of $\lambda$ values. The difference between LN and HN is minor compared to the dramatic improvement both provide over the unnormalized baseline. This confirms that addressing the scale mismatch through normalization is the key insight; the specific normalization technique is a secondary design choice.

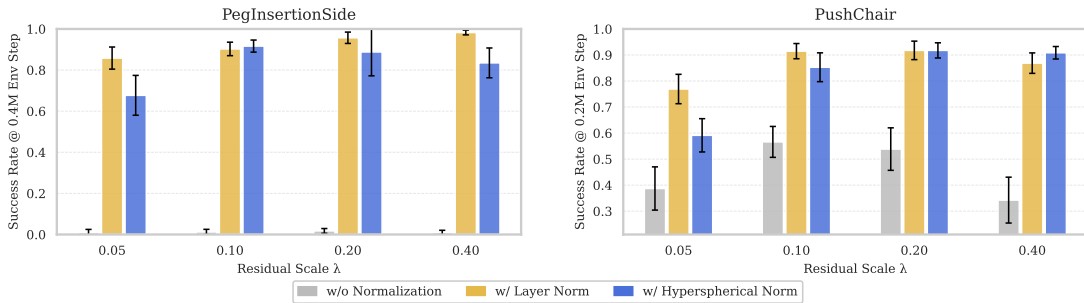

*Figure 19.* **Normalization is essential across residual scales.** Without normalization, the critic fails to learn efficiently regardless of $\lambda$. Both LN and HN substantially outperform the unnormalized baseline, with the gap between LN and HN being much smaller than the gap between either and no normalization.

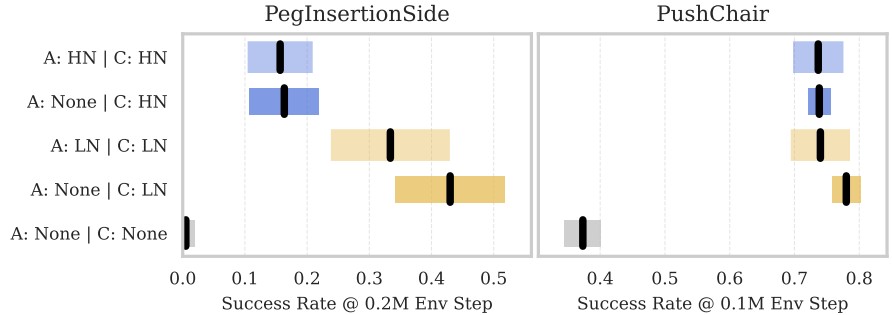

*Figure 20.* **Normalization benefits the critic but not the actor.** Adding normalization to the actor (A: LN or A: HN) provides no improvement over normalizing only the critic (A: None — C: LN or A: None — C: HN). The bottom row shows that without critic normalization, performance collapses regardless of actor normalization.

### D.1.2. WHY NORMALIZATION BENEFITS THE CRITIC BUT NOT THE ACTOR

Figure 20 presents a systematic ablation of normalization placement. We compare five configurations: no normalization, critic-only normalization (with LN or HN), and full normalization of both actor and critic.

The results reveal a clear pattern:

- **Critic normalization is necessary**: The bottom row (A: None — C: None) shows that without critic normalization, performance is severely degraded on both tasks.

- **Actor normalization is unnecessary**: Comparing "A: None — C: LN" with "A: LN — C: LN" shows that adding normalization to the actor provides no benefit and may slightly hurt performance on PegInsertionSide.

- **The pattern holds for both LN and HN**: The same trend appears with Hyperspherical Normalization.

This asymmetry confirms that scale mismatch is fundamentally a *value learning* problem. The critic takes the combined action $a = a_{\text{base}} + \lambda \cdot a_{\text{res}}$ as input, where the small $\lambda$ causes the residual component to be overshadowed by the base action. Normalization decouples the critic's internal representations from input magnitudes, restoring its ability to detect residual variations. The actor, in contrast, takes only the state as input and faces no such scale mismatch. Once the critic provides meaningful gradients, the actor can learn effectively without additional architectural modifications.

### D.1.3. DIAGNOSTIC METRICS

To understand *how* normalization improves value learning, we introduce two diagnostic metrics visualized in Figure 7.

**Critic Sensitivity.** We measure the critic's responsiveness to residual actions via the gradient norm:

$$\text{Sensitivity} = \mathbb{E}_{s \sim \mathcal{D}} \left[ \|\nabla_{a_{\text{res}}} Q_\theta(s, a_{\text{base}} + \lambda \cdot a_{\text{res}})\| \right] \tag{20}$$

This metric quantifies how strongly the critic's output responds to changes in the residual action. A higher sensitivity indicates that the critic can detect small variations in $a_\text{res}$, which is essential for providing meaningful policy gradients. Without normalization, the dominant magnitude of $a_\text{base}$ suppresses the critic's sensitivity to the residual component.

**Value Difference.** We measure the critic's ability to attribute value to the residual policy:

$$\text{Value Difference} = \mathbb{E}_{s\sim\mathcal{D}}\left[|Q_\theta(s, a_\text{base} + \lambda \cdot a_\text{res}) - Q_\theta(s, a_\text{base})|\right] \tag{21}$$

This metric captures the value contribution that the critic assigns to the residual action. As training progresses, a well-functioning critic should recognize that the residual improves upon the base policy and assign increasing value to it. Without normalization, this growth is substantially slower, indicating the critic struggles to distinguish the residual's contribution.

**Normalization Is Not Merely Gradient Rescaling.** A natural alternative explanation is that normalization simply compensates for the gradient suppression introduced by the small $\lambda$. Since the residual enters the action as $\lambda \cdot a_\text{res}$, the policy gradient satisfies $\nabla_{a_\text{res}} Q = \lambda \cdot \nabla_a Q$ and is scaled down by $\lambda$, so one might suspect that any mechanism restoring this magnitude would suffice and that normalization offers no advantage over directly rescaling the gradient. We test this directly by scaling the gradient at the residual action input by a factor $k \in \{1, 10, 20\}$, where $k = 10$ exactly compensates for $\lambda = 0.1$ and $k = 20$ over-compensates. This scaling affects only the actor: during critic training $a_\text{res}$ is a stored constant from the replay buffer, so critic weight updates are untouched and $k$ amplifies only the gradient that flows back to the policy. As shown in Figure 21, all three factors perform similarly to one another and remain substantially below Layer Norm on both tasks. Restoring, or even over-restoring, the gradient magnitude does not recover the benefit of normalization.

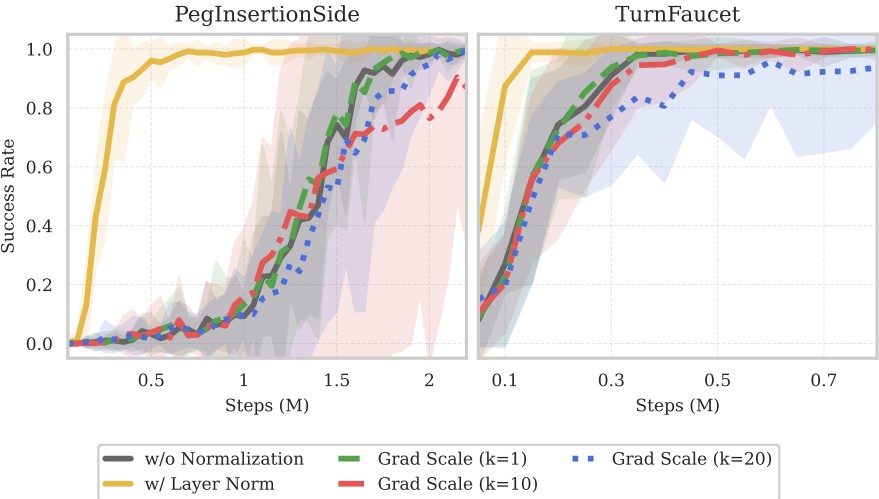

*Figure 21.* **Gradient scaling cannot substitute for critic normalization.** We ablate gradient scaling at the residual action input with $k \in \{1, 10, 20\}$, where $k{=}10$ provides the exact theoretical compensation for $\lambda{=}0.1$. Unlike Layer Norm, gradient scaling only amplifies the gradient passed to the actor during policy updates; it leaves critic weight updates unchanged since $a_\text{res}$ is a fixed constant during critic training. Despite this, all three scaling factors perform similarly to each other and substantially below Layer Norm on both tasks, ruling out insufficient actor gradient signal as the bottleneck and pointing to the critic's forward pass as the operative factor. Results on PEGINSERTIONSIDE and TURNFAUCET with 8 random seeds; shaded regions show 95 % CI.

This isolates the bottleneck. If the residual learned slowly merely because its policy gradient were too small, exact compensation ($k = 10$) would close the gap; it does not. The limitation instead lies in the critic's forward pass: when $a_\text{base}$ dominates the input magnitude, the unnormalized critic cannot represent the value difference induced by the residual in the first place, so there is no informative gradient for any $k$ to amplify. Two earlier observations corroborate this reading. First, the value-difference metric plateaus early without normalization (Figure 7) rather than growing with training, indicating that the critic never learns to attribute value to the residual rather than merely estimating it imprecisely. Second, normalizing the actor provides no benefit (Section D.1.2), confirming that the deficiency is specific to the critic's representation. Normalization helps because it acts on this representation, decoupling internal activations from input magnitude so that small residual variations remain distinguishable, an effect that gradient rescaling structurally cannot reproduce.

## D.2. Distributional Objectives

This section provides implementation details for the distributional RL experiments in Section 3.2.2.

### D.2.1. IMPLEMENTATION DETAILS

We compare the standard MSE objective against three distributional alternatives, all implemented within the SAC framework. The key modification for actor-critic integration is that the actor update uses the *expected value* (mean of the distribution) rather than the full distribution, while the critic learns the complete return distribution.

**C51.** Categorical distributional RL (Bellemare et al., 2017) models the return distribution as a categorical distribution over a fixed set of atoms. The critic outputs logits over $N$ atoms with fixed support $\{z_i\}_{i=1}^N$ uniformly spaced in $[V_{\min}, V_{\max}]$. The distributional Bellman target is projected onto this support, and the critic is trained with cross-entropy loss. For SAC integration, the entropy term is subtracted from the target values before projection:

$$\mathcal{T}Z(s,a) \overset{D}{=} r + \gamma \left( Z(s',a') - \alpha \log \pi(a' \mid s') \right) \tag{22}$$

The actor is updated using the expected Q-value: $\mathbb{E}[Z(s,a)] = \sum_i p_i z_i$.

Our experiments use a shifted sparse reward $r_t = r_t^{\text{sparse}} - 1 \in \{-1, 0\}$. For this setting, the theoretical value range is $V_{\min} = -(1 - \gamma^T)/(1 - \gamma)$ and $V_{\max} \approx 0$, where $T$ is the episode length. Table 7 lists the support ranges used for each task.

*Table 7.* C51 hyperparameters by task.

| Task | $\gamma$ | $T$ | $V_{\min}$ | $V_{\max}$ |
|---|---|---|---|---|
| PegInsertionSide | 0.97 | 200 | $-35$ | 0 |
| PushChair | 0.9 | 200 | $-10$ | 0 |

**QR-DQN.** Quantile Regression (Dabney et al., 2018) learns a set of return quantiles without requiring a fixed support. The critic outputs $N$ quantile estimates $\{\theta_i\}_{i=1}^N$ corresponding to quantile fractions $\tau_i = (2i - 1)/(2N)$. The loss is computed pairwise between all current quantiles and all target quantiles:

$$\mathcal{L}_{\text{QR}} = \frac{1}{N^2} \sum_{i=1}^N \sum_{j=1}^N \rho_{\tau_i}^\kappa \left( y_j - \theta_i(s,a) \right) \tag{23}$$

where $y_j = r + \gamma \theta_j(s', a') - \alpha \log \pi(a' \mid s')$ is the $j$-th target quantile, and $\rho_\tau^\kappa$ is the quantile Huber loss that weights overestimation and underestimation asymmetrically based on $\tau$. The actor uses the mean of the quantiles: $\bar{Q}(s,a) = \frac{1}{N} \sum_i \theta_i(s,a)$.

*Table 8.* QR-DQN hyperparameters.

| Hyperparameter | Value |
|---|---|
| Number of quantiles $N$ | 25 |
| Huber threshold $\kappa$ | 1.0 |

**TQC.** Truncated Quantile Critics (Kuznetsov et al., 2020) extends quantile regression with an ensemble of critics and truncates the highest quantile estimates to mitigate overestimation bias. When computing the target, we drop the top $d$ quantiles from each of the $K$ critics:

$$y = r + \gamma \left( \frac{1}{K(N - d)} \sum_{k=1}^K \sum_{i=1}^{N-d} \theta_{k,i}(s', a') - \alpha \log \pi(a' \mid s') \right) \tag{24}$$

where $\theta_{k,i}$ denotes the $i$-th lowest quantile from the $k$-th critic. This truncation provides a pessimistic value estimate that reduces the overestimation bias common in actor-critic methods.

*Table 9.* TQC hyperparameters.

| Hyperparameter | Value |
| --- | --- |
| Number of quantiles $N$ | 25 |
| Number of critics $K$ | 5 |
| Top quantiles dropped per critic $d$ | 2 |

### D.2.2. VALUE DISTRIBUTION ANALYSIS

Figure 10 in the main text visualizes the "anatomy" of residual value learning: despite the residual's small footprint in action space, its effect on Q-values manifests as a clear mean shift. Here we describe the experimental design and provide additional analysis across training.

**Setup.** We sample 1024 states from the replay buffer and compute two Q-values for each state:

- $Q(s, a_{\text{base}})$: the value of the base policy action alone.

- $Q(s, a_{\text{full}})$: the value of the combined action $a_{\text{base}} + \lambda \cdot a_{\text{res}}$.

We plot the distribution of these values across all sampled states.

**Scope of This Visualization.** We clarify what this analysis does and does not establish. The distributions in Figure 10 and Figure 22 aggregate a single scalar Q-value per state over many states; they are *not* the per-state return distributions $Z(s, a)$ that distributional critics model. We therefore do not use these figures to argue directly that the full return distribution is uninformative. The claim that distributional objectives offer no efficiency gain rests on the empirical comparison in Figure 9, where MSE matches or exceeds C51, QR, and TQC. What these figures support is the *mechanistic* reading of that result: the residual's effect on the critic's output, evaluated on the states the policy actually visits, takes the form of a stable mean shift in $Q$ rather than a change in distributional shape. Since MSE directly targets such mean differences, it is well-suited to capture the residual's value contribution, which offers a consistent explanation for why the added expressiveness of distributional critics yields no benefit here.

**Mean Shift Throughout Training.** The main text shows the Q-value anatomy at a single checkpoint. A natural question is whether this mean shift pattern persists throughout training or only appears at certain stages. Figure 22 tracks the Q-value distributions at five checkpoints from 200K to 1000K environment steps on PegInsertionSide.

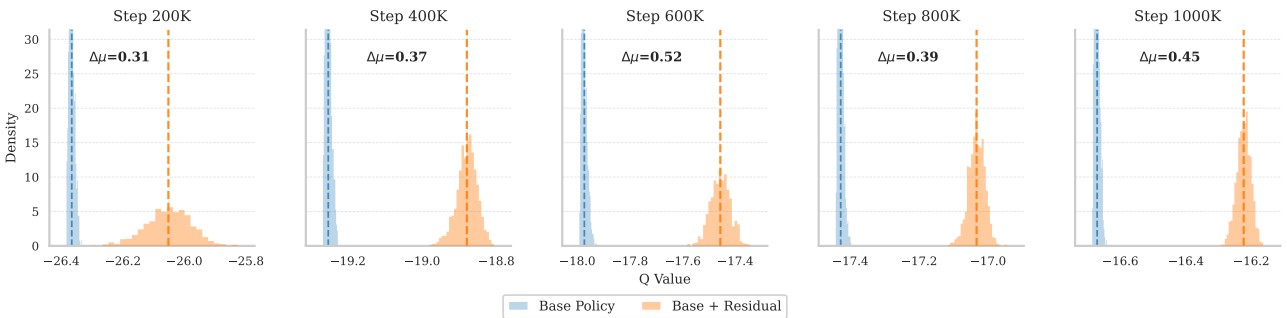

*Figure 22.* **Q-value distributions throughout training.** The mean shift between base policy (blue) and combined policy (orange) remains consistent across all training stages. As training progresses, overall Q-values increase (from ≈-26 to ≈-16), reflecting improved policy performance, while the separation $\Delta\mu$ between the two distributions remains stable (ranging from 0.31 to 0.52).

Two observations emerge from this analysis:

- **Consistent separation**: The mean shift $\Delta\mu$ between base and combined policy Q-values remains stable throughout training, ranging from 0.31 to 0.52. This indicates that the critic consistently recognizes the value contribution of the residual policy at all stages of learning.

- **Shape preservation**: The distribution shapes remain similar across checkpoints; the primary change is in location (mean) rather than spread or modality. This reinforces our finding that the residual's effect on value is well-characterized by a simple mean shift.

These results provide additional evidence that MSE-based critics are well-suited for residual value learning: since the value signal consistently manifests as a mean shift rather than complex distributional changes, directly optimizing for mean differences is both sufficient and efficient.

# E. `DAWN` Implementation

This section provides the complete algorithm and hyperparameters for `DAWN`.

## E.1. Algorithm

`DAWN` consists of two key modifications to the standard residual RL pipeline: (1) implicit warmup via base policy data collection, and (2) critic normalization using Layer Normalization. Algorithm 1 presents the complete training procedure.

---

**Algorithm 1 `DAWN`**: Data-Anchored Warmup and Normalization

---

**Require:** Base policy $\pi_{\text{base}}$, residual scale $\lambda$, warmup transitions $N_{\text{warmup}}$

1: Initialize residual policy $\pi_{\text{res}}$ with near-zero output
2: Initialize critic $Q_\theta$ with Layer Normalization          ◁ *Key modification*
3: Initialize target critic $Q_{\theta'} \leftarrow Q_\theta$, replay buffer $\mathcal{D} \leftarrow \emptyset$
4:
5: *// Phase 1: Value Anchoring*          ◁ *Key modification*
6: **while** $|\mathcal{D}| < N_{\text{warmup}}$ **do**
7:     $a_{\text{base}} \leftarrow \pi_{\text{base}}(s)$
8:     Execute $a = a_{\text{base}}$, observe $r, s'$          *// no residual, no training*
9:     Store $(s, a_{\text{base}}, r, s')$ in $\mathcal{D}$
10: **end while**
11:
12: *// Phase 2: Residual RL Training*
13: **for** each environment step **do**
14:     $a_{\text{base}} \leftarrow \pi_{\text{base}}(s)$
15:     $a_{\text{res}} \sim \pi_{\text{res}}(\cdot \mid s)$
16:     Execute $a = a_{\text{base}} + \lambda \cdot a_{\text{res}}$, observe $r, s'$
17:     Store $(s, a_{\text{base}}, a_{\text{res}}, r, s')$ in $\mathcal{D}$
18:
19:     **for** each gradient step **do**
20:         Sample mini-batch $\mathcal{B}$ from $\mathcal{D}$
21:         Update $Q_\theta$ by minimizing Bellman error (Eq. 7)
22:         Update $\pi_{\text{res}}$ by maximizing $\mathbb{E}[Q_\theta(s, a_{\text{base}} + \lambda \cdot a_{\text{res}}) - \alpha \log \pi_{\text{res}}]$
23:         Soft update: $Q_{\theta'} \leftarrow \tau Q_\theta + (1 - \tau) Q_{\theta'}$
24:     **end for**
25: **end for**

---

The key differences from prior residual RL methods are:

- **Warmup data uses base policy only**: Unlike exploration-based warmup strategies, we collect transitions using $\pi_{\text{base}}$ alone with zero residual actions. This grounds the critic to the true value landscape around the base policy.

- **No progressive exploration**: We omit the $\epsilon$-schedule used in Policy Decorator (Yuan et al., 2025). With efficient value learning via normalization, this protective mechanism becomes unnecessary.

- **Critic normalization**: The critic uses Layer Normalization to address scale mismatch, enabling it to detect small residual contributions.

- **No explicit critic pre-training**: Warmup data enters the buffer and the critic is trained jointly with $\pi_{\text{res}}$ from the first gradient step. **DAWN** has no policy-frozen pre-training phase, avoiding the entropy dominance that such a phase induces.

### E.2. Hyperparameters

To ensure fair comparison with Policy Decorator (Yuan et al., 2025), we adopt the same residual scale $\lambda$ for each task. Table 10 lists the complete hyperparameters across all experimental configurations.

*Table 10.* Residual scale $\lambda$ across all tasks and base policy configurations. Values are adopted from Policy Decorator for fair comparison.

| Task | Base Policy | $\lambda$ |
|---|---|---|
| *Diffusion Policy (state observations)* | | |
| PegInsertionSide | Diffusion Policy | 0.1 |
| TurnFaucet | Diffusion Policy | 0.1 |
| PushChair | Diffusion Policy | 0.2 |
| Pen | Diffusion Policy | 0.2 |
| Hammer | Diffusion Policy | 0.1 |
| Relocate | Diffusion Policy | 0.1 |
| *Diffusion Policy (visual observations)* | | |
| TurnFaucet | Diffusion Policy | 0.05 |
| PushChair | Diffusion Policy | 0.2 |
| *Behavior Transformer (state observations)* | | |
| Door | BeT | 0.3 |
| Pen | BeT | 0.3 |
| Hammer | BeT | 0.3 |
| Relocate | BeT | 0.2 |

Beyond the residual scale $\lambda$ inherited from Policy Decorator (Yuan et al., 2025), **DAWN** introduces only a single new hyperparameter: the number of warmup transitions $N_{\text{warmup}}$. Based on our investigation in Section 3.1.1, we use $N_{\text{warmup}} = 20$K across all tasks. This value provides sufficient data for value anchoring while adding negligible overhead to the total training budget.

**Integration with Base Policies.** **DAWN** is agnostic to the base policy architecture. Here we describe the integration details for the two base policy types used in our experiments.

- **Diffusion Policy**: Diffusion Policy (Chi et al., 2025) outputs a sequence of actions via iterative denoising. Following Policy Decorator, we use action chunking: the policy predicts 16 future actions, of which 4 are executed before re-planning. The residual policy outputs a corresponding sequence of 4 residual actions, which are added to the base actions element-wise after scaling by $\lambda$. The critic receives the summed action sequence flattened into a single vector.

- **Behavior Transformer**: BeT (Shafiullah et al., 2022) also uses action chunking with a context window. The integration follows the same pattern: the residual policy outputs actions matching the base policy's action horizon, and the critic evaluates the summed actions.

## F. Extended Experiments

This section provides the full results for the three robustness axes summarized in Section 5: comparison against stronger baselines (Appendix F.1), generality beyond SAC via a DDPG-based framework (Appendix F.2), and robustness to compute scaling (Appendix F.3).

### F.1. Comparison with Stronger Baselines

Our main experiments use Policy Decorator (Yuan et al., 2025) as the primary baseline. Here we situate **DAWN** against two additional families of methods: the concurrent residual RL method ResFiT (Ankile et al., 2025a), and recent non-residual approaches for fine-tuning generative policies.

**Comparison with ResFiT.** ResFiT (Ankile et al., 2025a) also seeds the replay buffer to address cold start, but its anchor is a *reward-annotated offline dataset*, whereas **DAWN** uses only on-policy rollouts from the base policy. To compare the two anchoring strategies on equal footing, we evaluate within the ResFiT setting on the Square task and isolate the data-anchoring component while keeping Layer Normalization fixed across all variants. Figure 23 compares three configurations: **DAWN** (base-policy rollouts as anchor), ResFiT (offline buffer as anchor), and a baseline with no data anchor. The unanchored baseline learns slowly, confirming that cold start is a genuine obstacle in this setting as well. **DAWN** matches the sample efficiency of ResFiT despite relying on a strictly weaker data assumption: it requires neither reward labels nor a curated offline dataset, only rollouts that the base policy can generate on demand. This supports our claim that grounding the critic to the base policy's value landscape, rather than the specific source of the anchoring data, is what resolves cold start.

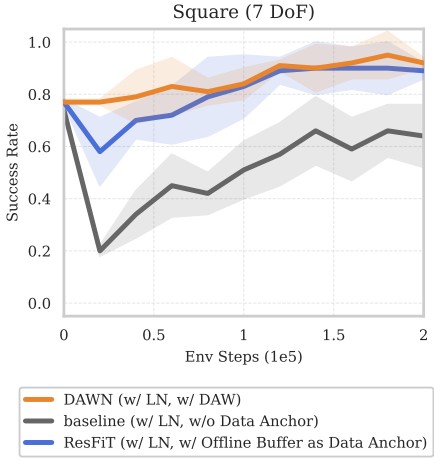

*Figure 23.* **DAWN matches ResFiT with a weaker data assumption.** On the Square task within the ResFiT setting, **DAWN** (base-policy rollouts as anchor) matches ResFiT (reward-annotated offline buffer as anchor), while the unanchored baseline learns slowly. **DAWN** achieves comparable efficiency without requiring reward labels or a curated offline dataset.

**Comparison with Non-Residual Fine-Tuning.** We further compare against DSRL (Wagenmaker et al., 2025) and DPPO (Ren et al., 2024), two recent methods that fine-tune generative policies without freezing the base. We evaluate on three representative tasks spanning both benchmarks (Adroit Hammer, Adroit Relocate, and ManiSkill PushChair).

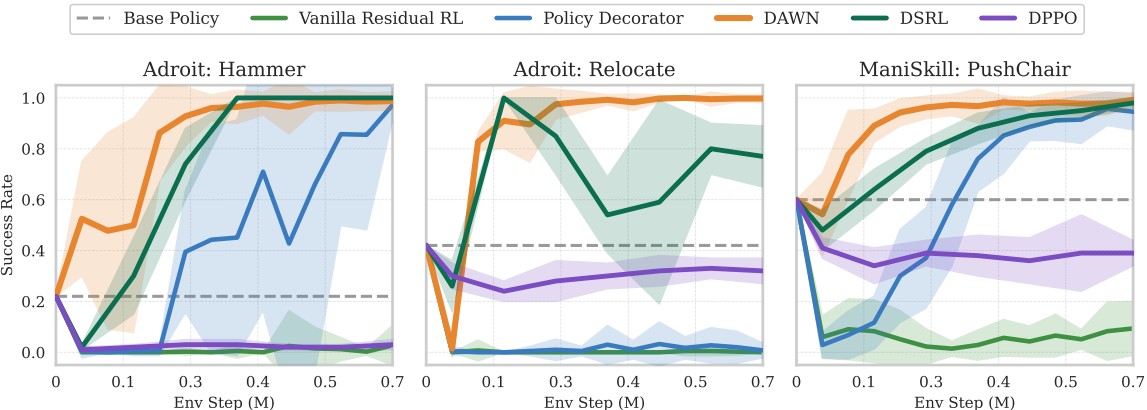

*Figure 24.* **DAWN compares favorably to non-residual fine-tuning methods.** We compare against DSRL and DPPO, two recent methods for fine-tuning generative policies via RL. Unlike **DAWN**, these methods do not freeze the base policy: DSRL optimizes over the diffusion policy's latent noise space, and DPPO directly fine-tunes the policy weights via policy gradient. This comparison is therefore not strictly apples-to-apples, as the methods differ in their assumptions about base policy access and the risk of catastrophic forgetting. Nevertheless, **DAWN** achieves competitive or superior sample efficiency across all three tasks while preserving the base policy intact, confirming that targeted improvements to residual value learning are an effective alternative to direct policy modification. Dashed lines indicate base policy performance. Results with 8 random seeds; shaded regions show 95 % CI.

## F.2. Generality Beyond SAC: `DAWN` in a DDPG-Based Framework

Our investigation in Section 3 uses SAC as the underlying algorithm, which raises a natural question: do cold start and scale mismatch arise from the residual formulation itself, or are they artifacts of SAC's entropy mechanism? To answer this, we instantiate `DAWN`'s two components within ResFiT, which is built on DDPG and therefore has no entropy term. These experiments use the Square task from the ResFiT evaluation setting, since that is the regime in which the DDPG-based framework is configured.

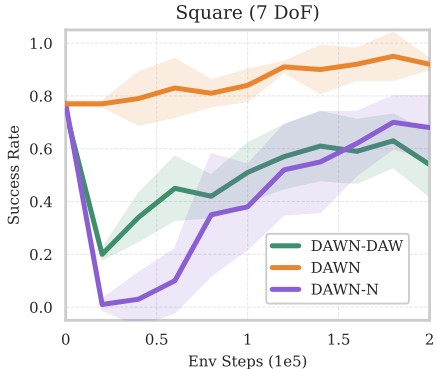

*Figure 25.* **Both `DAWN` components remain necessary in a DDPG-based framework.** Instantiated within ResFiT (DDPG-based) on the Square task, removing data-anchored warmup (DAWN-DAW) or normalization (DAWN-N) degrades performance, mirroring the SAC results in Figure 13. Since DDPG has no entropy term, this confirms that cold start and scale mismatch stem from the residual formulation rather than from SAC's entropy mechanism. Results with 8 random seeds; shaded regions show 95 % CI.

Figure 25 reports the component ablation in this framework: removing data-anchored warmup (DAWN-DAW) or critic normalization (DAWN-N) each degrades performance, mirroring the pattern observed under SAC. Because DDPG carries no entropy regularization, this result rules out SAC's entropy mechanism as the source of either bottleneck. Cold start persists because the critic is still randomly initialized and must be grounded to the base policy's value landscape; scale mismatch persists because the critic still receives the magnitude-dominated combined action $a_{\text{base}} + \lambda \cdot a_{\text{res}}$ as input. Both are structural consequences of the residual formulation, and both interventions remain necessary regardless of the algorithm.

## F.3. Robustness to Compute Scaling

A natural concern is whether `DAWN`'s gains could be matched simply by increasing compute rather than by addressing the underlying value-learning pathologies. We consider two axes of compute scaling on ManiSkill PushChair: the update-to-data (UTD) ratio, which controls how many critic updates are performed per environment step, and the critic ensemble size.

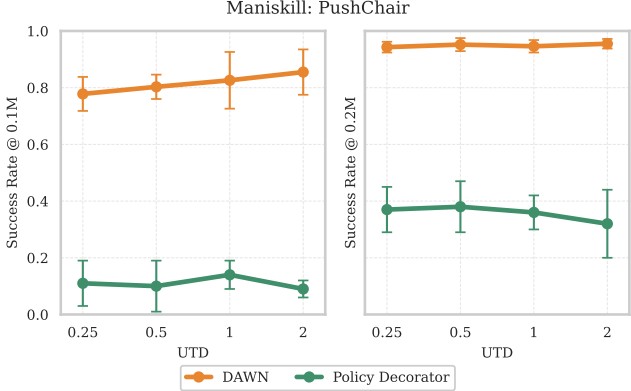

*Figure 26.* **`DAWN` outperforms Policy Decorator across all UTD ratios.** We ablate the update-to-data (UTD) ratio in $\{0.25, 0.5, 1, 2\}$ on ManiSkill PushChair, measuring success rate at 0.1M and 0.2M environment steps. Policy Decorator does not benefit meaningfully from higher UTD, while the gap to `DAWN` persists across the entire range.

**Update-to-Data Ratio.** We vary the UTD ratio for both **DAWN** and Policy Decorator and measure success rate at 0.1M and 0.2M environment steps. As shown in Figure 26, Policy Decorator does not benefit meaningfully from a higher UTD ratio, while the gap to **DAWN** persists across the entire range. This confirms that simply spending more compute on critic updates cannot substitute for resolving the underlying value-learning pathologies: when the critic is neither anchored nor sensitive to the residual, additional updates propagate an uninformative learning signal more often rather than a better one.

**Critic Ensembles.** A related concern is whether a larger critic ensemble could close the gap, since RLPD (Ball et al., 2023) and ResFiT use a 10-critic ensemble inheriting REDQ (Chen et al., 2021). Such ensembles reduce overestimation bias through output aggregation across critics, which is orthogonal to scale mismatch: every critic independently receives the same magnitude-dominated input $a_{\text{base}} + \lambda \cdot a_{\text{res}}$ and faces the same sensitivity failure. Figure 27 confirms this empirically. With normalization, **DAWN** (2 critics) and **DAWN**+REDQ (10 critics) perform identically despite the $5\times$ difference in critic count, while **DAWN** with normalization far exceeds **DAWN**+REDQ without it ($\approx 0.78$ vs. $\approx 0.42$ at UTD=0.25). The ensemble yields a modest gain only at UTD=2, where overestimation bias becomes relevant, so its $5\times$ overhead is unwarranted given that **DAWN** already reaches near-perfect success rates at default settings.

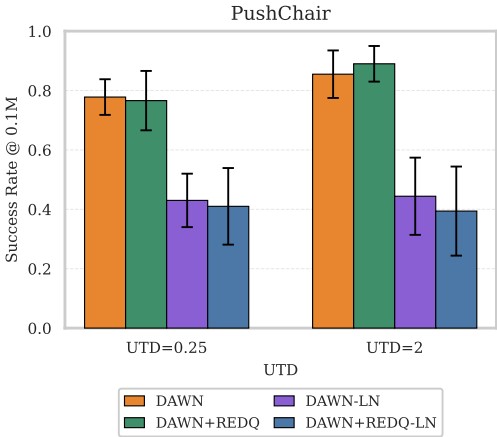

*Figure 27.* **Large critic ensembles cannot substitute for layer normalization. DAWN** (2 critics) vs. **DAWN**+REDQ (10 critics) and their no-LN counterparts (suffix indicating LN *removed*) at UTD $\in \{0.25, 2\}$ on ManiSkill PushChair. Removing LN degrades performance regardless of ensemble size, and **DAWN**+REDQ matches **DAWN** only when both are normalized.

# G. Additional Ablations

This section presents additional ablation studies to validate **DAWN**'s design choices.

## G.1. Sensitivity to Residual Scale

The residual scale $\lambda$ controls the magnitude of corrections the residual policy can apply. A natural concern is whether **DAWN** is sensitive to this hyperparameter. Figure 28 evaluates **DAWN** with two different $\lambda$ values against Policy Decorator on three ManiSkill tasks.

The results demonstrate that **DAWN** is robust to the choice of $\lambda$:

• On PegInsertionSide, both $\lambda$ values achieve superior efficiency, converging to 100% success rate before 1M steps.

• On TurnFaucet and PushChair, the performance gap between $\lambda = 0.1$ and $\lambda = 0.2$ is negligible.

• Across all tasks, both **DAWN** variants substantially outperform Policy Decorator in sample efficiency.

This robustness simplifies hyperparameter tuning in practice: practitioners can select $\lambda$ based on task-specific considerations (e.g., desired correction magnitude) without concerns about value learning stability.

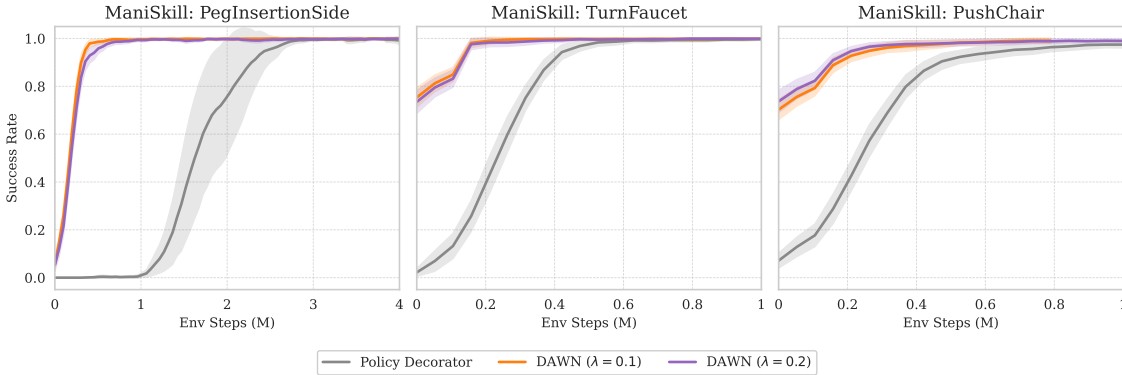

*Figure 28.* **Sensitivity to residual scale** $\lambda$. **DAWN** maintains strong performance across different $\lambda$ values (0.1 and 0.2), consistently outperforming Policy Decorator. The minimal gap between **DAWN** ($\lambda = 0.1$) and **DAWN** ($\lambda = 0.2$) indicates consistent robustness.

## G.2. Progressive Exploration is Unnecessary

Policy Decorator (Yuan et al., 2025) introduces progressive exploration, which gradually increases the probability of using the residual policy during training (Equation 6). This mechanism was designed to stabilize training when value learning is inefficient. We hypothesize that with efficient value learning via **DAWN**, this protective mechanism becomes unnecessary.

Figure 29 compares three configurations: (1) Policy Decorator without progressive exploration (our minimal baseline), (2) Policy Decorator with progressive exploration using the tuned schedule from Yuan et al. (2025), and (3) **DAWN** without progressive exploration. The progressive exploration schedules use $H = 30K$ for PegInsertionSide, $H = 100K$ for TurnFaucet, and $H = 300K$ for PushChair. The results reveal a clear pattern:

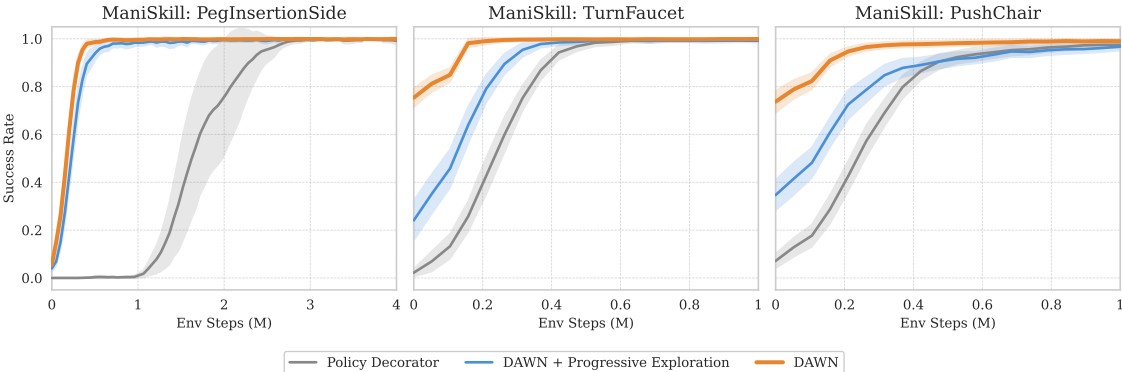

*Figure 29.* **Progressive exploration is unnecessary with efficient value learning.** **DAWN** without progressive exploration outperforms Policy Decorator even when the latter uses a carefully tuned progressive schedule. This confirms that addressing the root causes of inefficient value learning eliminates the need for protective heuristics.

- **Progressive exploration helps the baseline**: Adding progressive exploration to Policy Decorator improves performance, particularly on PegInsertionSide. This confirms its role as a protective mechanism against unstable value learning.

- **DAWN outperforms both**: Without any progressive exploration, **DAWN** achieves better sample efficiency than Policy Decorator with progressive exploration. This demonstrates that addressing the root causes (cold start and scale mismatch) is more effective than symptom-level fixes.

- **Simpler and more efficient**: **DAWN** eliminates the need for task-specific tuning of the schedule parameter $H$, which varies by an order of magnitude across tasks (30K to 300K in these experiments).

This ablation validates our design philosophy: rather than adding protective mechanisms to mask inefficient value learning, **DAWN** directly addresses the underlying pathologies, resulting in a simpler and more effective method.

