# OpenReview forum: "What Makes Value Learning Efficient in Residual Reinforcement Learning?"
_ICML.cc/2026/Conference — ICML 2026 spotlight_

### Official Review · Reviewer_39vk · 2026-03-04

**Soundness:** 2
**Presentation:** 3
**Significance:** 2
**Originality:** 1
**Overall Recommendation:** 4
**Confidence:** 3

**Summary:**

The paper tackles the problem of finetuning a pre-trained policy for robotic control with Residual Reinforcement Learning. The authors use an off-the-shelf off-policy RL algorithm (SAC) and identify two main problems that hinders the fine-tuning of the base policy, focusing on the learning of an accurate value function: (1) value function cold start: it seems that in the general setting, practitioners assume to have a base policy but not a base value function, the randomly-initialized value function being inaccurate around the base policy visitation space, this leads to early training instabilities (the authors propose a simple warmstarting technique to address that) (2) residual policy scaling issue:  the output action is a weighted sum of the base action and residual action, as the residual action weight is generally small, authors claim it hinders credit assignment and make the policy finetuning harder (they propose a simple layer normalization technique to address that).

The narrative of the paper is centered around proposing mechanistic explanations for why these problems happen and understanding how they could be addressed by testing and dissecting several principled solutions.

**Compliance With Llm Reviewing Policy:**

Affirmed.

**Final Justification:**

The authors clarified and addressed most of my concerns with additional explanations and plots. I thereby raise my score to weak accept. I will not champion the paper, but I'm okay with it being accepted.

**Key Questions For Authors:**

**On the warmstart part:**

1\. A part of the warmstart analysis focuses on an explicit warmup phase; in this phase the residual policy is frozen and only the critic is trained. It seems to me a lot of the problems faced are self-inflicted, due to an ill-initialized “near-deterministic” policy, leading to exploding entropy regularization. Did you try modifying the residual policy initialization in order to control better the entropy term?

2\. On the “why part” the explanations don’t click for me. You say that the failure of Hard-Q learning is due to objective mismatch, but you don’t specify how that makes it fail. Can you propose an explanation as to why the objective mismatch would deteriorate the value learning?

2 bis. You say that the explicit warmup is “redundant”. Can you clarify what this means in this context? Redundant with what?

**On the normalization part:**

3\. Can you provide evidence that the sensitivity w.r.t. the residual action is indeed what’s hindering the network from learning properly? You show that normalization improves the perf (Figure 6) and you show that during training you observed gradient magnitudes w.r.t. to the residual action about twice as big when using normalization (Figure 7). It could be that normalization just improves the overall accuracy of the value network, leading to naturally larger q-values and gradients, and there’s no need to bring up structural sensitivity to make this explanation plausible. A way to investigate sensitivity could be to try a gradient scaling layer directly on the residual action inputs.

**On the distributional RL part:**

Line 320-326, right column:

    >The left panel projects actions onto their first principal component. The residual introduces only a marginal shift in the action distribution, leaving the full policy nearly indistinguishable from the base in action space. This directly illustrates the scale mismatch challenge for value learning: the residual contribution is structurally suppressed, constituting only a bounded correction to the critic’s action input.

4\. I don’t understand this paragraph. Can the authors explain what “residual contribution is structurally suppressed” means? The only thing I see in the left part of Figure 10 is the fact that the contribution of the residual policy is smaller (by design) and that it makes the base policy shift a bit (this is intended, and I don’t identify the two policies to be indistinguishable).

4 bis. On the right panel of Figure 10, I don’t see why we’re looking at the distribution of Q-values across different states. If we are to comment on the utility of distributional RL, shouldn’t we look at the return distribution for a single state directly? Plotting the variance across the entire state-space conflates state-level variance with the actual return distribution. I can understand that you found experimentally that distributional RL worked worse in practice, but I don’t think there is enough evidence here to explain why that’s the case.



**Experiment details**

5\. What is the exact central tendency aggregation method you used for the plots (average, median, iqm)? While the captions mention 95% confidence intervals, the central metric itself should be explicitly specified.

5 bis. Also, the majority of ablations are conducted on a subset of the tasks, which is perfectly understandable, but can you be more upfront in the main text about what you actually looked at and why you chose those specific tasks for each of the figures?

5 ter. In Figure 13 (the main component ablation), why is the training stopped so early? It’s trained on about one half of the number of steps used in Figure 11. What about the asymptotics? Do the ablated-out baselines catch up in the long run?

**Limitations:**

The limitations are not addressed enough in my opinion, I couldn't find a limitations paragraph. I think overall the paper could be more upfront about what are evidence-backed claims and what are educated guesses, hypothesis or ad-hoc explanation.

**Strengths And Weaknesses:**

## Strengths
- The presentation of the paper is appealing; there are multiple polished figures, takeaways messages are highlighted, and this makes it a pleasant read.
- The paper’s intentions are laudable. I think identifying clear problems and coming up with explainable and simple solutions is an appealing line of work, and it could clearly benefit both the research community and the practitioners.
- The authors selected multiple simple baselines that a practitioner typically could have wanted to test. This brings a “we tested that for you so you don’t have to” spirit that is beneficial to the paper.
- The experiments were run on many tasks, and with a high number of random seeds (8). This reinforces the statistical significance of the evidence provided.

## Weaknesses

- I’m a bit worried about the weakness of the baselines. For instance, in Figure 11 the “vanilla residual RL” performs worse than the base policy on every single task, poorly tuned baseline, making it somewhat of a strawman comparison?

- There are a number of hasty conclusions and hand-wavy explanations that make it hard to fully agree with the claims of the paper. I believe that some claims, specifically regarding “why” these simple tricks and methods work, are not properly justified. I develop this point in the “Questions” section. Addressing correctly those concerns would make me upgrade my rating of the paper (see questions 1-4).

- The phrasing is not always clear. The term “bottleneck” is used a lot of times, sometimes to refer to value learning in general, sometimes to refer to the failure modes of the value learning. It is not indicated what the nature of the bottleneck is (I know what a speed or data bottleneck is, I don’t know what a value learning bottleneck is). Perhaps the terms “challenge” or “failure mode” are more appropriate. Also, the term “minimal” is somewhat over-used; the “policy decorator” is called a minimal baseline throughout the text, then, the “naive residual RL” is introduced in section 5 and it is also qualified minimal.

---

> ### Author Rebuttal · Authors · 2026-03-31
>
> We thank the reviewer for the constructive feedback. We address the technical questions in full below, and will incorporate the writing suggestions in the revised version.
>
> ---
> **[W1]** Vanilla residual RL performing below the base policy is not a tuning artifact but a direct manifestation of cold start pathology, and our implementation follows the Policy Decorator setup exactly. To strengthen the baseline comparison, we have added comparisons with ResFiT (DDPG-based) [**[Figure link]**](https://anonymous.4open.science/r/ICML-2026-Rebuttal-3B26/rebuttal_ReSFiT.pdf) and two representative expressive policy fine-tuning methods, DSRL and DPPO [**[Figure link]**](https://anonymous.4open.science/r/ICML-2026-Rebuttal-3B26/rebuttal_DSRL.pdf). DAWN achieves competitive or superior sample efficiency across all settings.
>
> ---
> [Q1] This question led us to correct an imprecise statement in our paper. Entropy dominance is an unavoidable structural consequence of freezing the policy, independent of initialization. The standard SAC parameterization ($L_{\min}{=}{-}20$, $L_{\max}{=}2$, following SpinningUp) gives $\mathbb{E}[\text{log-std}]=(L_{\min}{+}L_{\max})/2=-9$ for any zero-mean symmetric initialization, since $\mathbb{E}[\tanh(y)]=0$ holds exactly regardless of initialization scale. This yields $\alpha\cdot\mathbb{E}[|\log\pi|]\approx 7.58\,\alpha d\gg|r|\leq 1$, where $d$ is the effective action dimension including action chunking: ${\approx}2.12$ for TurnFaucet and ${\approx}6.06$ for PushChair.
>
> We verify this empirically by ablating fc_logstd std $\in \{0.01, 0.1, 1.0\}$ [**[Link]**](https://anonymous.4open.science/r/ICML-2026-Rebuttal-3B26/rebuttal_explicit.pdf). Across a $100{\times}$ range in std, $\mathbb{E}[\log\pi]$ remains constant at ${\approx}212.3$ (theoretical: $212.2$), and all variants collapse well below the MC return baseline. In standard SAC, actor updates immediately shift fc_logstd outputs away from zero, escaping this regime within the first gradient steps. Explicit warmup prevents this escape by keeping the policy frozen throughout pre-training.
>
> ---
> **[Q2]** Hard-Q pre-trains the critic on $r+\gamma\min Q'$, producing Q-values near the true MC return (e.g., $[-35, 0]$ on PegInsertionSide). When training switches to SAC, the per-step penalty $\alpha|\log\pi|\approx 2.12$ accumulates over the effective horizon $1/(1-\gamma)\approx 33$, shifting the steady-state target range to $\approx[-105, 0]$. The critic must refit to an entirely different value scale from scratch, making the switch point more costly than simply using random initialization with proper warmup data.
>
> ---
> **[Q3]** We agree that Figure 7 (left) alone does not rule out a general accuracy improvement, and will revise the mechanistic discussion accordingly. Three pieces of evidence collectively point to a critic-specific structural effect.
> - First, gradient scaling with $k \in \{1, 10, 20\}$ [[**Figure Link**]](https://anonymous.4open.science/r/ICML-2026-Rebuttal-3B26/rebuttal_gradient.pdf) rules out insufficient actor gradient signal: despite $k=10$ providing exact theoretical compensation for $\lambda=0.1$, all variants perform substantially below LN. Gradient scaling leaves critic weight updates unchanged, so this result points to the critic's forward pass as the key factor.
> - Second, without LN, $|Q_{\text{full}}-Q_{\text{base}}|$ grows only marginally and plateaus early (Figure 7, right), whereas a general accuracy improvement would be expected to produce sustained growth regardless.
> - Third, adding LN to the actor yields no benefit (Figure 14), confirming that the effect is critic-specific.
>
> ---
> **[Q4 & Q4bis]** We thank the reviewer for these precise observations, which help us clarify the intent of Figure 10.
>
> - On the left panel: we agree "nearly indistinguishable" overstates the overlap and will correct this. The left panel serves as a visual counterpart to the right: despite the small input-level shift, the critic with LN produces a clear Q-value mean separation ($\Delta\mu=0.27$), while without LN this attribution stagnates entirely, as shown by the flat $|Q_{\text{full}}-Q_{\text{base}}|$ curve in Figure 7 (right).
> - On the right panel: we agree that cross-state Q-value variance is not the same as a single-state return distribution, and our argument does not rely on the latter. The figure shows that the residual's value contribution manifests as a stable, systematic mean shift across 1024 diverse states (confirmed across training in Figure 21), which is precisely what MSE optimizes directly. Distributional RL must additionally model distribution shape, a strictly harder objective that offers no extra directional signal when the mean shift is already sufficient, naturally reducing sample efficiency as Figure 9 confirms.
>
> ---
> **[Q5]** All plots report the mean across seeds with 95% CI.
>
> **[Q5ter]** Full curves are at [**[Figure link]**](https://anonymous.4open.science/r/ICML-2026-Rebuttal-3B26/ablation_long.pdf).

---

> > ### Author Rebuttal · Reviewer_39vk · 2026-04-03
> >
> > I thank the authors for the answers. I fell like most of my concerns have been addressed. I strongly encourage the authors to incorporate the corrected text and additional experiments in their final manuscript. I will raise my score to weak accept following the rebuttal and answers to other reviewers.

---

> > > ### Author Response · Authors · 2026-04-08
> > >
> > > We thank the reviewer for the thorough engagement and for the encouraging acknowledgement. We wanted to gently flag that the final justification deadline is approaching, in case a submission is still pending. We will make sure all the discussed corrections and experiments are incorporated in the revised version.

---

### Official Review · Reviewer_qXQv · 2026-03-05

**Soundness:** 4
**Presentation:** 4
**Significance:** 3
**Originality:** 3
**Overall Recommendation:** 5
**Confidence:** 4

**Summary:**

This paper identifies and addresses two common issue that reduce the efficacy of residual reinforcement learning. They cover the "cold start pathology" and the "structural scale mismatch" problem, some intuitive fixes that surprisingly fail, other fixes that succeed, and combine the successful remediations together into their method that resolve these issues in residual RL. While not an algorithmically novel paper overall, this paper advances the science of residual RL by exhaustively investigating the why and the where for problems and their targeted solutions in residual RL. With their remediations and the experimental comparison to prior SOTA work, the authors clearly position their work as a necessary step in improving the pathologies that plague residual RL behind the scenes.

**Compliance With Llm Reviewing Policy:**

Affirmed.

**Final Justification:**

The authors responded to my questions well and have committed to adjusting the paper for camera ready to address my issue with related work not being covered in the main text. While the novelty is limited in building a "new" algorithm, their exhaustive empirical experiments to determine where and why issues break this style of algorithm is something the field of ML and RL sorely need more of. This is a really nice paper and I think it should be accepted.

**Key Questions For Authors:**

Why have the warmup data collection phase and critic regularization methods in this paper not previously been used in residual RL, especially for cases such as SAC? I don't think I have ever seen an online SAC implementation without warmup data from the policy (typically it is random policy since we are training from scratch, but still), so I am confused why this didn't get inherited by the residual RL community.

**Limitations:**

yes

**Strengths And Weaknesses:**

**Strengths:**
- The cadence and introduction of challenges and remediations throughout Section 3 is great and easy to follow.
- The usage of takeaway boxes further helps the reader follow the trajectory of the experiments and results.
- This paper reads as an investigation of a residual RL pathologies and succeeds in motivating and solving problems that have previously made these methods fail.
- This is a high compliment - this paper is just good science. While the novelty is limited in building a "new" algorithm, their exhaustive empirical experiments to determine where and why issues break this style of algorithm is something the field of ML and RL sorely need more of.
- I think this paper is especially timely as empirical papers hit a ceiling in terms of performance on many benchmarks. Other papers would simply inherit the use of a layer norm or hyperspherical normalization and not note their usage in the critic in their paper. I think papers such as this one that dig into the why for particular classes of problems are very important to allow the field to move forward from "mega algorithms" combining many "tricks" to instead favor algorithms with targeted "tricks" addressing known problems.

**Weaknesses:**
- As the authors state themselves, the power of Dawn is not in it's novel components. They certainly do a good job of motivating why and where these components are necessary.
- I haven't done an exhaustive literature review on this topic, but I think the lack of related work section in the main text makes it harder to determine novelty of this paper's work. For camera ready I would make room for the related work if at all possible or at least reference it in the main text more explicitly so the reader can better assess the field.

---

> ### Author Rebuttal · Authors · 2026-03-31
>
> We thank the reviewer for the thoughtful and generous assessment. We share the view that careful empirical investigation into ***why*** methods work or fail is just as valuable as proposing new algorithms, and we are glad this came through in the paper.
>
> We will move the related work coverage from the introduction into a dedicated section in the revised version to make the landscape easier to navigate.
>
> ---
> **[Question]** The short answer is that both techniques *were* inherited from standard SAC, but in a form that is insufficient for the residual RL setting. The deeper reason is that policy-side protective mechanisms systematically obscured the value-side failure modes, leaving them undiagnosed.
>
> **On warmup data collection:**
> - Standard off-policy implementations include a learning starts phase (typically 1K–10K steps; SpinningUp defaults to 10K, CleanRL to 5K), and this was carried over into residual RL unchanged. In standard from-scratch training this is reasonable: policy and value co-evolve from random initialization, so the quality and quantity of warmup data is largely inconsequential.
> - In residual RL the purpose is fundamentally different. The critic needs to anchor itself to the value landscape around an already-capable base policy before residual updates can be meaningful. Without sufficient base policy data, it actively misguides the residual policy from the first update, causing performance to collapse below the base policy before recovering. This recovery consumes a large portion of the sample budget, negating the primary advantage of starting from a strong prior.
> - The data composition is also silently wrong. Standard practice mixes in the randomly initialized residual policy during warmup, which introduces destructive perturbations for precision manipulation tasks. Figure 15 shows that base-policy-only collection consistently matches or outperforms any exploration-augmented strategy, counterintuitive from a standard RL perspective but consistent with the value anchor interpretation.
>
> **On critic normalization:**
> - The residual RL community has focused primarily on policy-side innovations: bounding residual magnitude, designing exploration schedules, managing catastrophic forgetting. This left value learning relatively unexamined, and recent advances in deep RL (SimBa, BRO) were not systematically applied to this setting.
> - More importantly, progressive exploration as in Policy Decorator gradually increases the residual policy's participation during training. This protects against unstable value learning, but in doing so prevents scale mismatch from ever manifesting clearly. Practitioners observing instability would reach for a more careful exploration schedule rather than questioning the critic's architectural sensitivity.
> - Concurrent work ResFiT does apply layer normalization inherited from RLPD, but applies it to the full network including the actor. Without mechanistic understanding, the right technique gets applied in the right place but also carried along unnecessarily elsewhere. Our analysis shows the effect is entirely critic-specific, because scale mismatch is an input representation problem localized to the critic.
>
> The two blind spots share a common origin: policy-side protective mechanisms masked value-side pathologies, making them difficult to identify without targeted investigation.
>
> ---
> We would also like to share several new experiments conducted during the rebuttal period.
>
> - **Baseline extension.** We added comparisons with ResFiT [[Link]](https://anonymous.4open.science/r/ICML-2026-Rebuttal-3B26/rebuttal_ReSFiT.pdf) and two representative diffusion policy fine-tuning methods, DSRL and DPPO [[Link]](https://anonymous.4open.science/r/ICML-2026-Rebuttal-3B26/rebuttal_DSRL.pdf). DAWN achieves competitive or superior sample efficiency across all settings without requiring a separately curated offline dataset, confirming that base policy rollouts collected online provide a sufficient value anchor.
> - **Generalization beyond SAC.** We validated DAWN within the DDPG-based ResFiT framework [[Link]](https://anonymous.4open.science/r/ICML-2026-Rebuttal-3B26/rebuttal_ReSFiT.pdf). Ablating either component substantially degrades performance under DDPG, confirming that cold start pathology and scale mismatch are intrinsic to the residual RL formulation rather than artifacts of SAC's entropy mechanism.
> - **UTD and ensemble ablations.** Higher UTD benefits DAWN consistently but leaves Policy Decorator unimproved [[Link]](https://anonymous.4open.science/r/ICML-2026-Rebuttal-3B26/rebuttal_UTD.pdf), showing that compute scaling cannot substitute for resolving the underlying value learning pathologies. A 10-critic ensemble likewise cannot substitute for layer normalization [[Link]](https://anonymous.4open.science/r/ICML-2026-Rebuttal-3B26/rebuttal_Ensemble.pdf): overestimation bias and scale mismatch act at entirely different levels, and a solution to one has no effect on the other.

---

> > ### Author Rebuttal · Reviewer_qXQv · 2026-04-03
> >
> > Thanks to the authors for their response to my questions and willingness to move the related work to the main text from the appendix. I have the same opinion as I did before and think this paper should be accepted. I will leave my score as is.

---

> > > ### Author Response · Authors · 2026-04-08
> > >
> > > We thank the reviewer for the encouraging assessment and for the kind words. We will make sure the related work section is well integrated in the revised version.

---

### Official Review · Reviewer_hhnk · 2026-03-10

**Soundness:** 3
**Presentation:** 4
**Significance:** 3
**Originality:** 2
**Overall Recommendation:** 5
**Confidence:** 4

**Summary:**

This paper investigates why value learning is inefficient in residual RL, where a pretrained base policy is refined by learning a small residual correction. The authors identify two key bottlenecks: cold start pathology, where the critic initially lacks value knowledge near the base policy, and structural scale mismatch, where the residual action is too small for the critic to properly attribute value. To address these issues, they propose DAWN, which seeds training with warmup transitions from the base policy, and applies normalization to the critic network.

**Compliance With Llm Reviewing Policy:**

Affirmed.

**Final Justification:**

The rebuttal addressed most of my concerns, and so I'm raising my score

**Key Questions For Authors:**

See my weakness section. I am very much willing to raise my score once those have been addressed.

**Limitations:**

DAWN seems to solve all the tasks in the paper quite easily. It would be nice to try harder tasks and see how well DAWN performs, and identify the bottlenecks.

**Strengths And Weaknesses:**

strengths:
- Residual RL is of great interest to the community, and it is important to understand what are the bottlenecks in residual RL. I think this work provides an important mechanistic understanding that benefits the community.
- Presentation is great! The bottlenecks of residual RL are well explained, and the analysis and experiment carry the flow well. The takeaway sections also make it crystal clear the main ideas of the paper. The plots are also well made and easy to read.
- The analysis and experiment are clear, and clearly demonstrate the usefulness of warmup and normalization.
- While the ideas presented in the paper are not original, the authors make a worthwhile contribution by analyzing their usefulness in residual RL.

weaknesses:
- Lack of discussion of relevant work: since the ideas presented in the paper are not original, it is important to discuss the line to work that presented those ideas, and clearly discuss the similarities and differences. For example, the warmup phase is introduced in [1], although in a non residual RL setting. They also discuss value collapse and the ineffectiveness of explicit warmup phase. There are very relevant to the main sections of the paper, yet the authors have no discussion of this. [2] also uses a variant of such warmup phase.
- Lack of extensive baselines: the authors show that DAWN work very well compared to two baseline methods, vanilla residual RL and policy decorator. However, these baselines are rather limited and there are more methods that ought to be compared to, such as [2], [3], or others.
- Comparison to non-residual RL that adopts warmup phase and critic normalization: while these methods are not an apples-to-apples comparison, it would be nice to see the comparison of residual RL to non-residual RL with the same tricks.

[1] Efficient Online Reinforcement Learning Fine-Tuning Need Not Retain Offline Data
[2] Residual Off-Policy RL for Finetuning Behavior Cloning Policies
[3] Stable Reinforcement Learning with Expressive Policies

---

> ### Author Rebuttal · Authors · 2026-03-31
>
> **W1:** We thank the reviewer for this feedback. We will add a dedicated related work section. Below we clarify the key connections and distinctions.
>
> **WSRL.** Both works share a common design philosophy: on-policy transitions from a capable policy provide meaningful signal during the transition to online RL. However, they address different problems. WSRL operates in offline-to-online fine-tuning, where a pre-trained Q-function already exists and warmup data recalibrates it across the distribution shift. Our DAW addresses cold start in residual RL, where the critic is randomly initialized and must be anchored to the base policy's value landscape from scratch.
>
> We also note a slight inaccuracy in the review: [1] demonstrates that *retaining offline data* is unnecessary, not that explicit critic pre-training fails. In fact, explicit value warmup is effective in standard offline-to-online settings [4], precisely because those policies are not near-deterministic and therefore do not suffer from the entropy dominance unique to residual RL.
>
> [4] Efficient Online RL Fine Tuning with Offline Pre-trained Policy Only
>
> **ResFiT.** ResFiT operates within residual RL and uses warmup data from reward-annotated offline demonstrations.
> - DAW requires only base policy rollouts collected online at the start of training, with no need for a separately curated offline dataset or reward annotation.
> - Despite this, DAWN matches or exceeds ResFiT in sample efficiency (see W2 response for results).
>
> ---
> **W2:** We thank the reviewer for this suggestion and have added comparisons to ResFiT [2] and two representative methods for fine-tuning expressive generative policies.
>
> **Comparison with ResFiT [[Figure Link]](https://anonymous.4open.science/r/ICML-2026-Rebuttal-3B26/rebuttal_ReSFiT.pdf).** ResFiT is the most directly comparable residual RL baseline. A key distinction lies in data requirements: ResFiT seeds the replay buffer with a separately curated offline dataset with reward annotations, whereas DAWN requires only base policy rollouts collected online at the start of training. Despite this, DAWN matches or exceeds ResFiT in sample efficiency on Square (7 DoF) (Figure 1). Figure 2 further shows that both DAWN components remain essential within the DDPG-based ResFiT framework: removing either data-anchored warmup or critic normalization substantially degrades performance, confirming that cold start pathology and structural scale mismatch arise from the residual RL formulation itself rather than being artifacts of SAC, and that DAWN's solutions generalize across off-policy algorithms.
>
> **Comparison with expressive policy fine-tuning methods [[Figure Link]](https://anonymous.4open.science/r/ICML-2026-Rebuttal-3B26/rebuttal_DSRL.pdf).** We compare against DSRL [5] (CoRL 2025) and DPPO [6] (ICLR 2025). This is not a strictly apples-to-apples comparison: DSRL steers the diffusion policy through its latent noise space, and DPPO directly fine-tunes policy weights—both under different assumptions regarding base policy access and catastrophic forgetting. Nevertheless, DAWN achieves competitive or superior sample efficiency across all three tasks while keeping the base policy intact, confirming that targeted improvements to residual value learning are an effective alternative to direct policy modification.
>
> [5] Steering Your Diffusion Policy with Latent Space Reinforcement Learning
>
> [6] Diffusion Policy Policy Optimization
>
> ---
> **W3:** We agree this comparison is valuable. Direct SAC fine-tuning of Diffusion Policy is technically infeasible since the multi-step denoising chain makes log π analytically intractable, so we use Gaussian MLP policies as non-residual counterparts under identical conditions.
>
> Results at 0.2M steps are shown below. Residual RL benefits roughly 5× more from warmup and LN than non-residual methods: +0.433/+0.602/+0.650 vs. near-zero gains for BC→SAC and SAC from scratch on the hardest task, and +0.202/+0.079 and +0.144/+0.110 on the remaining two. This asymmetry confirms that **DAW and LN address structural bottlenecks specific to residual RL rather than only serving as general-purpose improvements.** The absolute performance gap further reflects the advantage of preserving the Diffusion Policy prior: even with the same tricks, neither non-residual method approaches the performance ceiling that residual RL achieves.
>
> |Task|Method|w/o tricks|w/ tricks|Δ|
> |-|-|-|-|-|
> |**PegInsertionSide**|Residual RL|0.000 ± 0.00|0.433 ± 0.05|**+0.433**|
> ||BC→SAC|0.000 ± 0.00|0.000 ± 0.00|+0.000|
> ||SAC from scratch|0.000 ± 0.00|0.000 ± 0.00| +0.000 |
> |**TurnFaucet**|Residual RL|0.388 ± 0.18|0.990 ± 0.02|**+0.602**|
> ||BC→SAC|0.211 ± 0.10|0.413 ± 0.05|+0.202|
> ||SAC from scratch|0.103 ± 0.02|0.182 ± 0.04|+0.079|
> |**PushChair**|Residual RL|0.293 ± 0.10|0.943 ± 0.02|**+0.650**|
> ||BC→SAC|0.201 ± 0.12|0.345 ± 0.17|+0.144|
> ||SAC from scratch|0.032 ± 0.03|0.142 ± 0.05|+0.110|
>
> *Success rate at 0.2M steps (mean ± 95% CI, 8 seeds).*

---

> > ### Author Rebuttal · Reviewer_hhnk · 2026-04-01
> >
> > Thanks for the authors' response. This addresses most of my concerns and I have raised my score accordingly. I have one final suggestion: in your response to W3, you have compared DAW to Gaussian SAC policies. However this is not quite a fair comparison because your are comparing flow policies to Gaussian policies. I would encourage the authors to add a comparison to a flow-based SAC policy (e.g. FQL [7] or others) in the camera ready version.
> >
> > [7] Flow Q Learning

---

> > > ### Author Response · Authors · 2026-04-08
> > >
> > > We thank the reviewer for raising the score and for this helpful suggestion. We agree that a flow-based non-residual baseline such as FQL would make the W3 comparison more architecturally consistent. We are actively running this experiment and will include the results in the revised version.

---

### Official Review · Reviewer_2ttS · 2026-03-18

**Soundness:** 4
**Presentation:** 4
**Significance:** 3
**Originality:** 3
**Overall Recommendation:** 5
**Confidence:** 4

**Summary:**

This paper investigates the challenges of value learning in residual RL, where, unlike in standard RL, a frozen base policy is used together with a small residual actor-critic. The authors identify two key bottlenecks: (1) cold-start pathology, and (2) scale mismatch, where the critic struggles to determine the contribution of the residual action relative to the base action. Based on several analyses, they propose DAWN, which combines warm-up trajectory collection with critic-layer normalization to address these two bottlenecks. In their experiments, DAWN outperformed the baselines on the ManiSkill and Adroit benchmarks across two different base policy classes.

**Compliance With Llm Reviewing Policy:**

Affirmed.

**Final Justification:**

The rebuttal addressed my concerns, and I think this paper should be accepted.

**Key Questions For Authors:**

- Have the authors tried to ablate the update-to-date (UTD) ratio? Can using a higher UTD ratio further improve sample efficiency?


- I am also curious if increasing critic ensemble size can help resolve the structural scale mismatch. For example, the RLPD paper shows that using a 10-critic ensemble can significantly improve the sample efficiency. Does this also hold in the residual RL setting?

**Limitations:**

Yes

**Strengths And Weaknesses:**

**Stengths**

The paper is well written and provides many insights into residual RL. While the resulting approach is simple (warmup + layer normalization), there are extensive analyses to justify why these designs help. The decomposition of the two bottlenecks, cold-start pathology and scale mismatch, provides interesting findings regarding residual RL. The additional ablations regarding explicit value warmup and the different entropy variants provide valuable insights. The experimental results are strong compared to naive residual RL and policy decorator.

 **Weaknesses**

**[Major]** The evaluation is limited to simulated benchmarks. Given that the paper motivates the method for making residual RL efficient for real-world deployment, I would recommend adding real-world experimental results.

**[Major]** The experiments are conducted with relatively small base policies (diffusion policy and BeT). Similar to the point above, I am curious whether these results hold for larger VLAs, such as pi0 and OpenVLA.

 **[Major]** While the paper provides a lot of design decisions that are useful to improve SAC-based Residual RL, it is unclear if these takeaways are generalizable to other off-policy RL algorithms. It would be valuable to add an ablation over different algorithms to see if the conclusion holds.

 **[Minor]** The novelty of the individual components for the proposed method is limited. Although I can see that the paper’s contribution is primarily analytical rather than methodological, I am raising this as a minor weakness here.

---

> ### Author Rebuttal · Authors · 2026-03-31
>
> **[Major 1 & 2]** We agree these are important directions for future work. Within the residual RL literature, concurrent works have already demonstrated strong practical applicability: ResFiT [1] shows residual RL on real-world high-DoF humanoid systems, and recent work applies it to VLA fine-tuning [2]. Our contribution is complementary: the bottlenecks we identify arise from the residual RL formulation itself and are not specific to any particular base policy architecture or deployment environment, making the analysis broadly applicable as the paradigm scales to these settings.
>
> As additional supporting evidence for Major 2 specifically, we have added comparisons against DSRL (CoRL 2025 Oral) and DPPO (ICLR 2025) [[**Figure Link**]](https://anonymous.4open.science/r/ICML-2026-Rebuttal-3B26/rebuttal_DSRL.pdf), two of the most representative methods for fine-tuning diffusion policies in real-world applications. DAWN achieves competitive or superior sample efficiency across all three tasks, providing indirect evidence that our value learning improvements translate to settings where these methods are already deployed in practice.
>
> ---
> **[Major 3]** We address this directly with new experiments in the DDPG-based ResFiT [1] framework [[**Figure Link**]](https://anonymous.4open.science/r/ICML-2026-Rebuttal-3B26/rebuttal_ReSFiT.pdf). Two findings confirm generalization beyond SAC. First, DAWN matches or exceeds ResFiT despite requiring strictly less: base policy rollouts collected online at the start of training provide a sufficient value anchor, whereas ResFiT relies on a separately curated offline dataset with reward annotations. Second, ablating either DAWN component within this DDPG framework substantially degrades performance, confirming that both bottlenecks are intrinsic to the residual RL formulation rather than artifacts of SAC. Together, these results support the generality of our analysis across off-policy algorithms.
>
> ---
> **[Q1]** Yes, we ablated UTD $\in \{0.25,0.5,1,2\}$ on ManiSkill PushChair [[**Figure Link**]](https://anonymous.4open.science/r/ICML-2026-Rebuttal-3B26/rebuttal_UTD.pdf).
>
> DAWN benefits modestly from higher UTD early in training (0.78→0.86 at 0.1M steps), but converges to near-perfect performance across all UTD ratios by 0.2M steps. Policy Decorator shows no improvement from higher UTD at either point, and performs slightly worse at UTD=2 by 0.2M steps. When the underlying value learning pathologies are unresolved, increasing update frequency only applies a poorly anchored and insensitive critic more aggressively.
>
> The quantitative contrast is telling. At 0.1M steps, DAWN at the default UTD=0.25 already achieves 0.78, while Policy Decorator at UTD=2 reaches only 0.09. Scaling compute by 8$\times$ yields a gain of +0.08 for DAWN; addressing the value learning bottlenecks yields a gain of +0.69 over Policy Decorator at the same budget. Higher UTD and better value learning are therefore complementary rather than interchangeable: UTD controls how often the critic is updated, while DAWN determines whether those updates are grounded in an accurate value landscape and sensitive to residual contributions.
>
> ---
> **[Q2]** RLPD's 10-critic ensemble inherits the design of REDQ [3], which stabilizes high-UTD training by reducing overestimation bias through random ensemble subsampling. This operates at the level of output aggregation across critics, and is therefore orthogonal to scale mismatch, which is an input representation problem: every critic in the ensemble independently receives the same magnitude-dominated input and faces the same sensitivity failure. A solution to one cannot substitute for a solution to the other.
>
> We verify this directly by comparing DAWN (2 critics) against DAWN+REDQ (10 critics) and their no-LN counterparts at UTD $\in \{0.25, 2\}$ [[**Figure Link**]](https://anonymous.4open.science/r/ICML-2026-Rebuttal-3B26/rebuttal_Ensemble.pdf). The key comparison is between DAWN-LN and DAWN+REDQ-LN: despite a 5$\times$ difference in critic count, both configurations perform identically at both UTD ratios, confirming that ensemble size alone cannot resolve scale mismatch. Accordingly, DAWN with 2 critics and LN (≈0.78) substantially outperforms DAWN+REDQ-LN with 10 critics and no LN (≈0.42) at UTD=0.25.
>
> At the default UTD=0.25, DAWN+REDQ provides no benefit over DAWN. At UTD=2, a modest gain appears, consistent with ensembles mitigating overestimation bias at high update frequencies—precisely the problem REDQ was designed to solve, and one that is orthogonal to scale mismatch. Given that DAWN already achieves near-perfect success rates at default settings, the 5$\times$ compute overhead is not warranted in the residual RL setting.
>
> ---
> [1] Residual Off-Policy RL for Finetuning Behavior Cloning Policies
>
> [2] Unified Latent Steering and Residual Refinement for Online Improvement of Diffusion Policy Models
>
> [3] Randomized Ensembled Double Q-Learning: Learning Fast Without a Model

---

> > ### Author Rebuttal · Reviewer_2ttS · 2026-04-05
> >
> > I thank the authors for their response and the additional ablations. The response resolved most of my concerns, hence I have raised my score.

---

> > > ### Author Response · Authors · 2026-04-08
> > >
> > > We thank the reviewer for the careful reading, the constructive feedback, and for raising the score. We are glad the additional experiments and clarifications were helpful, and we will incorporate all the discussed improvements in the revised version.

---

### Decision · Program_Chairs · 2026-04-30

**Decision:**

Accept (spotlight)

**Comment:**

All reviewers strongly recommend the paper. The paper demonstrates a strong and systematical empirical study, which unveils clear insights and gives simple solution. Based on the reviews, I believe this paper will be impactful to the community of residual RL.